# Learning noise-induced transitions by multi-scaling reservoir computing

Zequn Lin [1,2,3,4,6], Zhaofan Lu[2,6], Zengru Di [2] & Ying Tang [1,2,5] ✉

Noise is usually regarded as adversarial to extracting effective dynamics from time series, such that conventional approaches usually aim at learning dynamics by mitigating the noisy effect. However, noise can have a functional role in driving transitions between stable states underlying many stochastic dynamics. We find that leveraging a machine learning model, reservoir computing, can learn noise-induced transitions. We propose a concise training protocol with a focus on a pivotal hyperparameter controlling the time scale. The approach is widely applicable, including a bistable system with white noise or colored noise, where it generates accurate statistics of transition time for white noise and specific transition time for colored noise. Instead, the conventional approaches such as SINDy and the recurrent neural network do not faithfully capture stochastic transitions even for the case of white noise. The present approach is also aware of asymmetry of the bistable potential, rotational dynamics caused by non-detailed balance, and transitions in multi-stable systems. For the experimental data of protein folding, it learns statistics of transition time between folded states, enabling us to characterize transition dynamics from a small dataset. The results portend the exploration of extending the prevailing approaches in learning dynamics from noisy time series.

Noise-induced transitions are ubiquitous in nature and occur in diverse systems with multi-stable states[1]. Examples include switches between different voltage and current states in the circuit[2], noisy genetic switches[3], noise-induced biological homochirality of early life self-replicators[4], protein conformational transitions[5,6], and chemical reactions[7] with the multi-stable probability distribution[8]. Learning noise-induced transitions is vital for understanding the critical phenomena of these systems. In many scenarios, only time series are available without mathematical equations known in prior. To effectively learn and predict noise-induced transitions from time series, it is necessary to distinguish both slow and fast time scales: fast relaxation around distinct stable states and slow transitions between them, where

the fast time-scale signals are often referred to noise[9,10]. Consequently, it remains elusive to learn stochastic transitions from time series in general.

Recently, many efforts have been made to learn the dynamics from data by machine-learning methods[11–20]. One type of approach uses Sparse Identification of Nonlinear Dynamics (SINDy) for identifying nonlinear dynamics, denoising time-series data, and parameterizing the noisy probability distribution from data[21,22]. Due to the nonconvexity of the optimization problem, the method may struggle to robustly handle large function libraries for the regression. Another type of approach employs physics-informed neural networks for data-driven solutions and discoveries of partial differential equations[23–25], or

[1]Institute of Fundamental and Frontier Sciences, University of Electronic Science and Technology of China, Chengdu 611731, China. [2]Department of Systems Science, Faculty of Arts and Sciences, Beijing Normal University, Zhuhai 519087, China. [3]Center for Interdisciplinary Studies, Westlake University, Hangzhou 310024, China. [4]School of Science, Westlake University, Hangzhou 310024, China. [5]Key Laboratory of Quantum Physics and Photonic Quantum Information, Ministry of Education, University of Electronic Science and Technology of China, Chengdu 611731, China. [6]These authors contributed equally: Zequn Lin, Zhaofan Lu. ✉e-mail: jamestang23@gmail.com

Koopman eigenfunctions from data[26]. However, the method requires an extensive quantity of data to train the deep neural network and refinements of the network.

Despite the broad application of the aforementioned methods, to our knowledge, they have not been utilized in studying noise-induced transitions. To learn noise-induced transitions, we first utilize the SINDy[22] and recurrent neural network (RNN)[27] for data with noise. We find that SINDy and RNN do not faithfully predict stochastic transitions, even for a one-dimensional bistable system with Gaussian white noise. We also apply the filters[28,29] to the data, obtain the smoothed time series, and then deal with the filtered data by SINDy[21]. Still, this method does not accurately capture noise-induced transitions. Similarly, the method of First-Order, Reduced, and Controlled Error (FORCE) learning[30], including its various versions of full-FORCE and the spiking neuron model[31], does not fully capture stochastic transitions in the experimental data and requires relatively high computational cost. These attempts indicate that these conventional methods were mainly designed for denoising the noisy data to learn the deterministic dynamics, rather than capturing the noise-induced phenomena. We thus need to develop a new approach to predict stochastic transitions.

We notice that one machine-learning architecture, reservoir computing (RC)[11,17], may be suitable for this task. The training of reservoir computing only needs linear regression, which is less computationally expensive than the neural network that requires back-propagation. Reservoir computing was found effective for learning dynamical systems[12,32,33], including chaotic systems[34–38]. A recent research started to apply reservoir computing to stochastic resonance[39], however, the functional role of noise in shifting dynamics between stable states has not been investigated. Another attempt employed RC for noise-induced transitions[40] but relied on an impractical assumption of knowing the deterministic dynamics equation beforehand. In practice, prior knowledge of deterministic dynamics is often lacking and sometimes even cannot be directly described by an equation[6]. Thus, can we forecast noise-induced transitions solely based on data without any prior knowledge of the underlying equation?

In this study, we develop a framework of multi-scaling reservoir computing for learning noise-induced transitions in a model-free manner. The present method is inspired by that the hyperparameter $\alpha$ in the reservoir was found to determine the time scale of reservoir dynamics[41]. Given a multi-scale time series, we can thus tune the hyperparameter $\alpha$ to match the slowly time-scale dynamics. After the reservoir captures the slowly time-scale dynamics by fitting the output layer matrix, we can separate the fast time-scale series as a noise distribution. During the predicting phase, we utilize the trained reservoir computer to simulate the slowly time-scale dynamics, and then add back the noise sampled from the separated noise distribution (for white noise) or learnt from the second reservoir (for colored noise). The whole protocol is iterated overtime points as the rolling prediction. Notably, the present method is different from previous work that regards noise merely as a disturbance[22,42], and instead focuses on capturing noise-induced transitions from the data.

To demonstrate the effectiveness of the present method, we apply it to two categories of scenarios. One type has the data generated from stochastic differential equations (SDE) for the purpose of testing the method, and the other has the experimental data[6]. For the first category with white noise, it includes a one-dimensional (1D) bistable gradient system, two-dimensional (2D) bistable gradient and non-gradient systems[43], 1D and 2D gradient systems with a tilted potential, a 2D tilted non-gradient system, and a 2D tristable system[44]. The present approach can capture statistics of the transition time and the number of transitions. For the first category with colored noise, we study a 1D bistable gradient system with Lorenz noise (Lorenz-63 model and Lorenz-96 model[40]), and accurately predict the specific transition time, without the assumption of knowing the deterministic

part of dynamics as required in[40]. For the second category, we apply the approach to the protein folding data[6], and explore the least amount of required data for accurate training, which can help reduce the demand for extensive measurements in experiments.

## Results

### The problem and conventional approaches

To study noise-induced transitions, we consider two types of data: one type is generated from SDE and the other type is experimental data. First, we use data generated from SDE. The continuous-state and continuous-time Markovian stochastic dynamics can be given as

$$\dot{\mathbf{u}} = f(\mathbf{u}) + \sigma\xi(t), \tag{1}$$

where the vector $\dot{\mathbf{u}}$ is the time derivative, the deterministic part of the dynamics is $f(\mathbf{u})$, and $\sigma$ corresponds to the noise strength. The $\xi(t)$ is a $k$-dimensional Gaussian white noise with $\langle\xi(t)\rangle = 0, \langle\xi(t)\xi^\top(t')\rangle = \delta(t-t')I_k$, where $\top$ denotes the transpose, $I_k$ is the $k$-dimensional identity matrix, $\delta(t-t')$ is the Dirac $\delta$ function, and $\langle\cdots\rangle$ represents the average.

Recent methods for learning dynamical models from time series have not directly handled the stochastic transitions (Supplementary Fig. 1). We apply the methods related to our work for the example below. First, we utilize two types of SINDy, SINDy-2021[22] and SINDy-2016[21,45] (Supplementary Fig. 2, Supplementary Table 1). The SINDy-2021 can learn the dynamics from data with noise and separate the noise distribution. However, it does not faithfully find stable states or predict stochastic transitions, while requiring high computational cost (Supplementary Table 2). The SINDy-2016 also does not capture the stochastic transitions from data with noise. The second method is RNN[27,46], which still does not accurately predict the stochastic transitions (Supplementary Fig. 3). Besides, the SINDy-2016 is not designed for data with noise. We thus preprocess the data by filters (Kalman filter[28] and Savizky-Golay filter[29]). However, SINDy-2016 still does not predict the noise-induced transitions for the filtered data (Supplementary Fig. 4, Supplementary Table 3). Moreover, we find that the FORCE learning method[31] can capture transitions in the bistable system with white noise (Supplementary Figs. 5, 6), but requires higher computational cost. It also does not faithfully learn stochastic transitions from the experimental data (Supplementary Figs. 7, 8).

### The framework of multi-scaling reservoir computing

Given that previous approaches are not applicable to noise-induced transitions, we leverage reservoir computing to learn the transitions. In reservoir computing[12,34], the input layer of the reservoir transforms time series into the reservoir network, while the output layer transforms the variables of the reservoir back to time series. The output layer is trained to minimize the difference between the input and output, with tuning the hyperparameters. It has the following scheme:

$$\mathbf{r}_{t+1} = (1-\alpha)\mathbf{r}_t + \alpha\tanh(A\mathbf{r}_t + W_{in}\mathbf{u}_t), \tag{2}$$

$$\tilde{\mathbf{u}}_{t+1} = W_{out}\mathbf{r}_{t+1}. \tag{3}$$

Here, the vector $\mathbf{u}$ is an $n$-dimensional state vector, and the initial condition is $\mathbf{u}_0$ with the lower script denoting the time, the $W_{in}$ is the input matrix with the values uniformly sampled in $[-K_{in}, K_{in}]$, the $\mathbf{r}$ is the $N$-dimensional reservoir state vector, the $A$ is the adjacency matrix of an Erdős-Rényi network with average degree $D$ to describe the reservoir connection between $N$ nodes, and the $\rho$ is the spectral radius of $A$. The tanh represents our activation function for this study. The $\bar{\mathbf{u}}$ is the output vector and $W_{out}$ is the output matrix. The $\alpha$ is the leak hyperparameter, representing the time scale[41], which becomes clearer

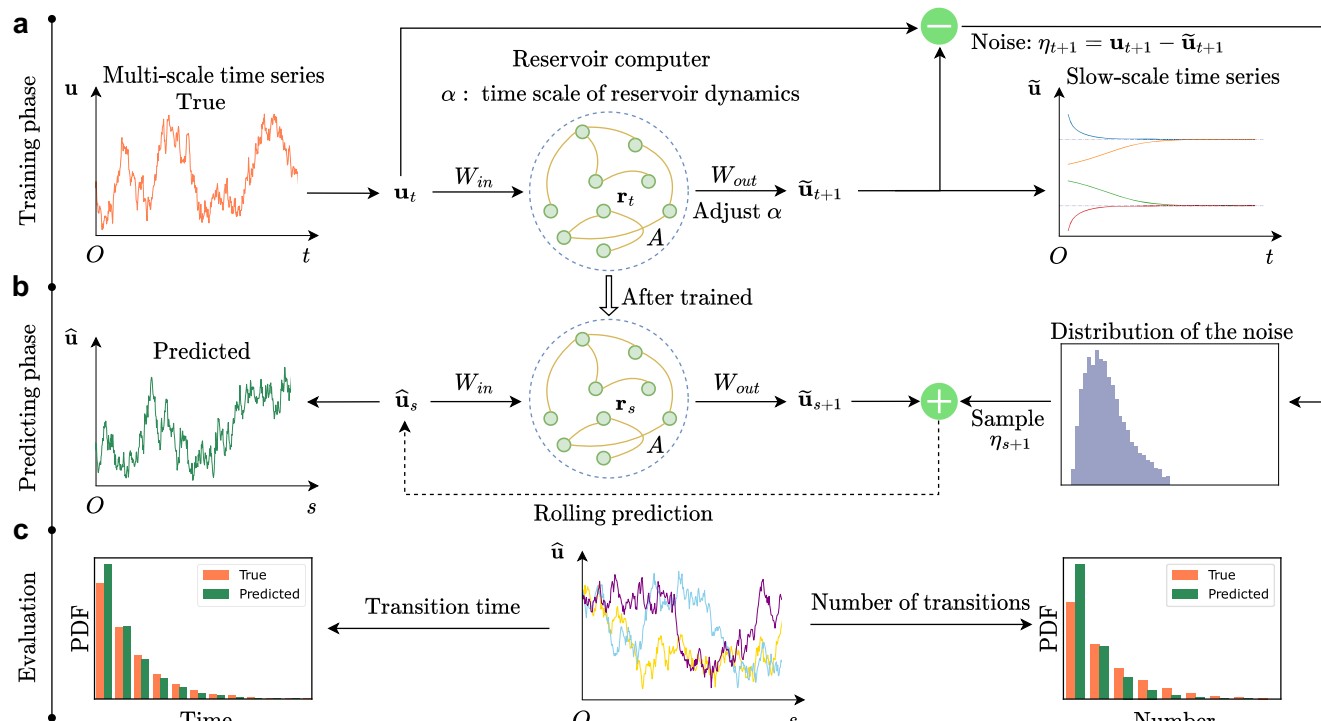

**Fig. 1 | Framework of learning noise-induced transitions by multi-scaling reservoir computing. a** The training data is a time series **u** with slow and fast time scales, and the fast time-scale part can be considered as noise, causing noise-induced transitions between stable states. In the training phase, at each time step $t$, the reservoir takes into $\mathbf{u}_t$ through a matrix $W_{in}$ and has a reservoir state $\mathbf{r}_t$ with a connection matrix $A$. The output matrix $W_{out}$ is trained to fit the output time series to the training data at the next time point. Tuning the hyperparameter $\alpha$ alters the time scale of the output $\tilde{\mathbf{u}}$, and a properly chosen $\alpha$ leads to a match with the slowly time-scale data. Then, $\mathbf{u} - \tilde{\mathbf{u}}$ separates the fast time-scale signal $\boldsymbol{\eta}$ as a noise distribution. **b** In the predicting phase, the $\hat{\mathbf{u}}_s$ is put into the trained reservoir to generate $\tilde{\mathbf{u}}_{s+1}$. In the next time step $s+1$, the input $\hat{\mathbf{u}}_{s+1}$ is the $\tilde{\mathbf{u}}_{s+1}$ plus the noise $\boldsymbol{\eta}_{s+1}$ sampled from the separated noise distribution. This process is iterated as a rolling prediction[56] to generate the time series $\hat{\mathbf{u}}$. **c** The evaluation on the predicted transition statistics. In the middle, different colored lines of $\hat{\mathbf{u}}$ represent replicates of the predictions. The accuracy is evaluated by the statistics of transition time and the number of transitions. PDF: probability density function.

when we rewrite Eq. (2) in its continuous-time form:

$$\frac{1}{\alpha}\dot{\mathbf{r}} = -\mathbf{r} + \tanh(A\mathbf{r} + W_{in}\mathbf{u}). \tag{4}$$

In the training phase, only $W_{out}$ is trained to minimize the difference between the output time series and the training data[47]. With the regularization term, the loss function is given by

$$L = \sum_{t=1}^{T} ||\mathbf{u}_t - W_{out}\mathbf{r}_t||^2 + \beta||W_{out}||^2, \tag{5}$$

where $\beta$ is the regression hyperparameter. We then regress the matrix $W_{out}$ by minimizing the loss function (Methods). By stacking the vectors of different time points as a vector: $\mathbf{U} \doteq [\mathbf{u}_1, ..., \mathbf{u}_T]$ and $\mathbf{R} \doteq [\mathbf{r}_1, ..., \mathbf{r}_T]$ with $t = 1, ..., T$, it can be rewritten as a compact form:

$$W_{out} = (\mathbf{UR}^\top) \cdot (\mathbf{RR}^\top + \beta)^{-1}. \tag{6}$$

Determining $W_{out}$ is a simple linear regression, which is less computationally expensive than the neural network that requires the back propagation.

The framework for learning noise-induced transitions using multi-scaling reservoir computing is summarized in Fig. 1. A reservoir acquires a time series **u** that contains signals with both fast and slow time scales. Given that $\alpha$ characterizes the time scale of reservoir computer[41], we search for an appropriate value of $\alpha$ to capture the slow time scale initially. After identifying an appropriate $\alpha$ value, additional

searches are conducted to find suitable values for other hyperparameters. This process aims to improve the accuracy of the results and obtain the trained slow-scale model. We utilize the trained slow-scale model to separate the noise distribution from the original time series. Then, we sample noise from the separated distribution and employ the trained slow-scale model for rolling prediction.

We use trial and error to search for appropriate hyperparameters[47]. In detail, the first strategy employs the information of stable states in the training set, which can be inferred by segmenting the time series between large jumps and calculating the mean value of each segment[6]. If reservoir computing effectively captures the slowly time-scale dynamics, the generated trajectories from various initial points (e.g., ten chosen points) should converge to the corresponding stable state. To achieve that, we tune the hyperparameter $\alpha$ and then refine the remaining hyperparameters. In the case of nonconvergence, the hyperparameter is adjusted in the opposite direction. When the hyperparameter adjustments do not further improve the convergence, we turn to the next hyperparameter[47]. The second strategy does not rely on prior information of stable states, where the hyperparameters are searched by evaluating the power spectral density (PSD). The accuracy of learning deterministic dynamics is quantified by the match of PSDs between the predicted time series and the training data (Supplementary Fig. 9). Thus, the match between the PSDs serves another indicator on the proper choice of hyperparameters.

After finding the appropriate hyperparameters, we utilize the trained slow-scale model to separate the noise distribution. Within the training phase, at time step $t$, the reservoir accepts the input $\mathbf{u}_t$, resulting in an output $\tilde{\mathbf{u}}_{t+1}$. Then, the noise at time step $t + 1$ can be

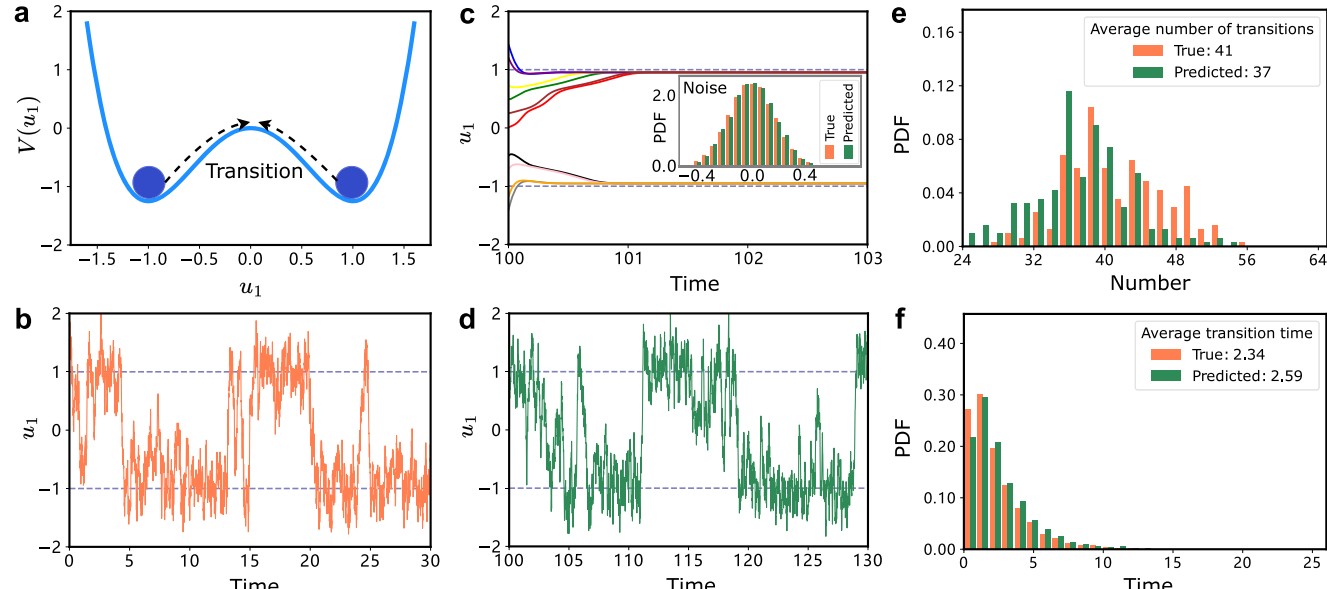

**Fig. 2 | Capturing stochastic transitions in a bistable gradient system with white noise. a** Schematic of noise-induced transitions in the bistable gradient system with Gaussian white noise. **b** Generated time series from Eq. (9) ($b = 5$, $c = 0$, $\varepsilon = 0.3$, $u_1(0) = 1.5$, $\delta t = 0.01$) with $t = 30$ as the ground truth. **c** The trained slow-scale model transforms ten different start points into ten different slowly time-scale series (color lines), and the noise distribution is separated. **d** The

prediction for $t \in [100, 130]$. **e** The number of transitions for the test and predicted data matches. Transition refers to the shift from $u_1 = -1$ to $u_1 = 1$ or vice versa. The duration of the prediction is $10000\delta t$. **f** Histograms of transition time for the test and predicted data. Transition time refers to the interval between two consecutive transitions.

computed as

$$\boldsymbol{\eta}_{t+1} = \mathbf{u}_{t+1} - \tilde{\mathbf{u}}_{t+1}. \tag{7}$$

We then obtain the noisy time series and the distribution as depicted in Fig. 1a. We can continue to implement the rolling prediction (Fig. 1b). In the predicting phase, at time step $s$, the reservoir accepts $\hat{\mathbf{u}}_s$, yielding the output $\tilde{\mathbf{u}}_{s+1}$. By adding $\boldsymbol{\eta}_{s+1}$, sampled from the noise distribution, to the output $\tilde{\mathbf{u}}_{s+1}$ as

$$\hat{\mathbf{u}}_{s+1} = \tilde{\mathbf{u}}_{s+1} + \boldsymbol{\eta}_{s+1}, \tag{8}$$

the $\hat{\mathbf{u}}_{s+1}$ is used as the input for the time step $s + 1$. The vector $\hat{\mathbf{u}}$ is the prediction.

As illustrated in Fig. 1c, to validate that the present method accurately captures noise-induced transitions, we compare the prediction with the test data. For white noise that is memoryless, we quantify the accuracy of the prediction by the statistics of noise-induced transitions. Instead of predicting a single transition, we focus on learning the statistics of transition time and the number of transitions from a set of trajectories. For colored noise, we aim to accurately forecast the occurrence of a specific noise-induced transition.

We next proceed with two categories of examples. One category is data generated from stochastic differential equations, including a 1D bistable gradient system and a 2D bistable non-gradient system with white noise, as well as a 1D bistable gradient system with colored noise. More examples are provided in Supplementary Note: a 1D tilted bistable gradient system (Supplementary Fig. 10, Supplementary Table 4), a 2D bistable gradient system (Supplementary Fig. 11, Supplementary Table 5), 2D tilted bistable gradient (Supplementary Fig. 12, Supplementary Table 6) and non-gradient (Supplementary Fig. 13, Supplementary Table 6) systems, a 2D tristable system (Supplementary Fig. 14, Supplementary Table 7), and a 1D bistable system with high-dimensional colored noise (Supplementary Fig. 15, Supplementary Table 8). The second category focuses on experimental data, where we apply the present method to protein folding data[6]. We also assess the

performance of using a small part of the dataset (Supplementary Fig. 16).

## Examples

**A bistable gradient system with white noise.** As a first example, we consider a 1D bistable gradient system with white noise[9]:

$$\dot{u}_1 = -b(-u_1 + u_1^3 + c) + \sqrt{2\varepsilon b}\xi_1(t), \quad t \ge 0. \tag{9}$$

The $\xi_1(t)$ is a Gaussian white noise. The parameter $b$ denotes the strength of the diffusion coefficient, $\varepsilon$ is the noise strength, and $c$ controls the tilt of the two potential wells. The system has noise-induced transitions between the two potential wells as illustrated in Fig. 2a. We generated a time series lasting $20000\delta t$, with the training set $t \in [0, 100]$ and the predicting set $t \in [100, 200]$. Figure 2b shows the first $3000\delta t$ of the training set.

In the training phase, the tuning of hyperparameters for the slow-scale model is performed as the protocol in our framework. After finding the proper hyperparameters listed in Table 1 (Example 1), we obtain the trained slow-scale model and the separated noise distribution (Fig. 2c). We remark that the convergence speed of the captured deterministic dynamics may have discrepancies compared with the actual dynamics. As a result, the separated noise distribution may exhibit lower or higher intensity compared with the actual noise distribution. In this case, we can employ a factor to magnify or reduce the noise strength. For instance, we amplify the sampled noise by a factor of 1.1 here, which improves the accuracy of the predictions (Supplementary Fig. 17).

In the evaluation, we conduct rolling prediction as shown in Fig. 1. The first $3000\delta t$ of the prediction is illustrated in Fig. 2d. The prediction has similar noise-induced transition dynamics to the test data. Next, we generate 100 replicates of time series from Eq. (9), train the model, and produce 100 time series separately. We then compare the statistics of the noise-induced transitions for these two sets of time series, e.g., the number of transitions over $10000\delta t$ (Fig. 2e) and the transition time (Fig. 2f). The match between the test and predicted

**Table 1 | The list of hyperparameters used for the different examples of the main text**

| Model | $\delta t$ | $T_{train}$ (Time steps) | $T_{predict}$ (Time steps) | $N$ | $K_{in}$ | $D$ | $\rho$ | $\alpha$ | $\beta$ |
|---|---|---|---|---|---|---|---|---|---|
| Example 1 | 0.01 | 10000 | 10000 | 800 | 4 | 4 | $1.2 \times 10^{-3}$ | 0.2 | $1 \times 10^{-8}$ |
| Example 2 set 1 | 0.01 | 8000 | 8000 | 1000 | 1.5 | 3.2 | $1.3 \times 10^{-3}$ | 0.25 | $1 \times 10^{-7}$ |
| Example 2 set 2 | 0.01 | 580 | 300 | 800 | 0.996 | 0.996 | 0.806 | 0.065 | $1 \times 10^{-7}$ |
| Example 3 | 0.002 | 20000 | 20000 | 1200 | 1 | 2.2 | $1.7 \times 10^{-3}$ | 0.3 | $1 \times 10^{-7}$ |
| Example 4 | real data | 25000 | 100000 | 800 | 0.04 | 1.8 | 0.021 | 0.22 | $1 \times 10^{-6}$ |

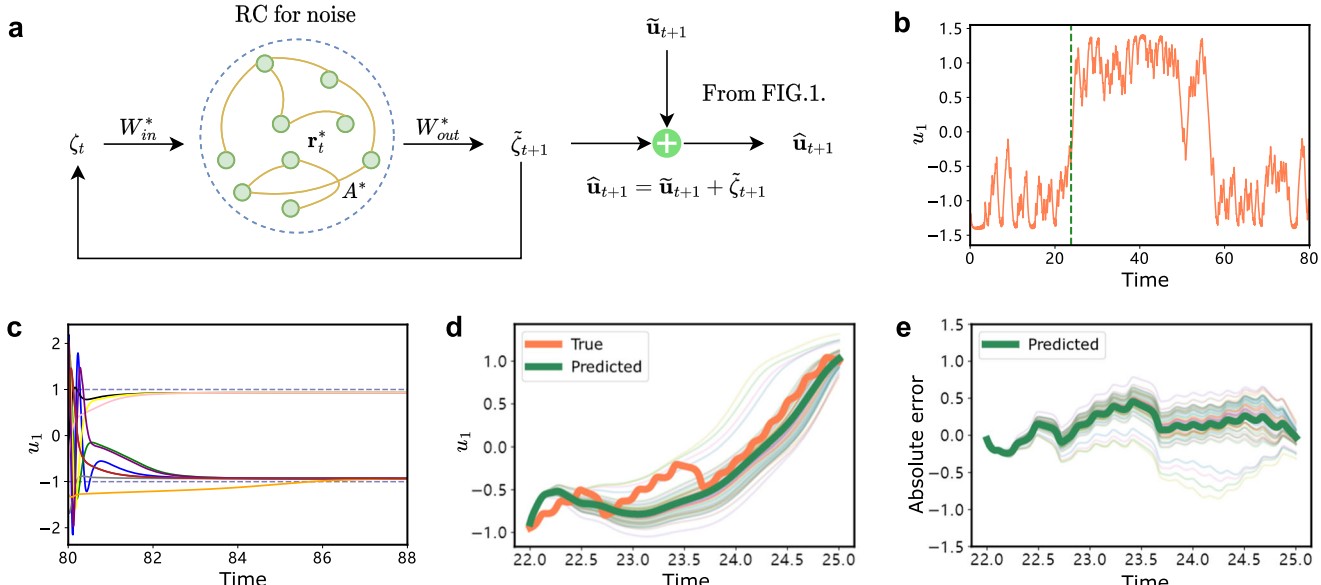

**Fig. 3 | Predicting the accurate transition time for a bistable gradient system with colored noise.** The system is the same as Eq. (10)[40]. **a** The flowchart of predicting stochastic transitions with colored noise. The process for obtaining noise $\zeta_t$ follows that in Fig. 1a, and a second reservoir takes into $\zeta_t$ through matrix $W_{in}^*$ and has reservoir states $\mathbf{r}_t^*$ with a connection matrix $A^*$. The output matrix $W_{out}^*$ is trained to learn noise. **b** Target time series ($x(0) = y(0) = z(0) = 1$, $b = 1$, $c = 0$, $\psi = 0.08$, $\epsilon = 0.5$, $u_1(0) = -1.5$, $\delta t = 0.01$) with $8000\delta t$, where a noise-induced transition occurs in $t \in [22, 25]$ marked by the green dashed line. The noisy data

from $580\delta t$ (with a range of $550\delta t$ to $650\delta t$ empirically suitable) before the stochastic transition at $t = 22$ is applied to predict the noisy time series in $t \in [22, 25]$. **c** The trained slow-scale model transforms ten different start points into ten different slowly time-scale series (color lines). **d** By repeating the process in **a** with the same hyperparameters, 50 predicted $u_1(t)$ are obtained (fainter lines). The averaged predicted time series (thick green) matches the test data (coral). **e** Absolute error of the predicted 50 time series and its mean value (thick green).

data demonstrates the effectiveness of our approach in capturing noise-induced transition dynamics.

**A bistable gradient system with colored noise.** The prediction on a single stochastic transition becomes possible when the system has colored noise. We need to learn the time evolution of the separated noise. Since RC is good at learning deterministic system, we employ a second RC (the first RC for the deterministic part) to learn the noise series for predicting a single transition. To demonstrate that the present method is applicable to such cases, we consider a system Eqs. (10) to (13) studied in ref. 40, where their method relies on the assumption of knowing the deterministic part of the equation in prior. In contrast, we do not assume any prior knowledge of the deterministic part of the dynamical system and directly learn both the deterministic part and noise (Fig. 3a), enabling prediction in a model-free manner.

The system is a 1D bistable gradient system, as illustrated in Fig. 3b:

$$\dot{u}_1 = -b(-u_1 + u_1^3 + c) + \frac{\psi}{\epsilon} y, \qquad (10)$$

$$\dot{x} = \frac{10}{\epsilon^2}(y - x), \qquad (11)$$

$$\dot{y} = \frac{1}{\epsilon^2}(28x - xz - y), \qquad (12)$$

$$\dot{z} = \frac{1}{\epsilon^2}\left(xy - \frac{8}{3}z\right). \qquad (13)$$

The parameter $b$ denotes the strength of the diffusion coefficient, $\epsilon$ corresponds to the noise strength, $\psi$ controls the influence of the noise on the slow-scale dynamics, and $c$ controls the tilt of the two potential wells. The noise $(x, y, z)$ is modeled by the Lorenz-63 model[48]. The system has stochastic transitions between the two potential wells under the Lorenz noise.

To test the present method, we consider the time series with a stochastic transition prior to the green dashed line in Fig. 3b. In the training phase, we obtain a slow-scale model to learn the deterministic part (Fig. 3c) and to separate noise. The hyperparameters are listed in Table 1 (Example 2 set 1). In the predicting phase, accurately forecasting the stochastic transitions requires predicting the noisy time series. Thus, we utilize a second reservoir (Fig. 3a) to learn the previously separated noise during the training phase. The hyperparameters for the noisy time series are listed in Table 1 (Example 2 set 2). With the deterministic component slow-scale model, we execute a rolling prediction to predict a single transition.

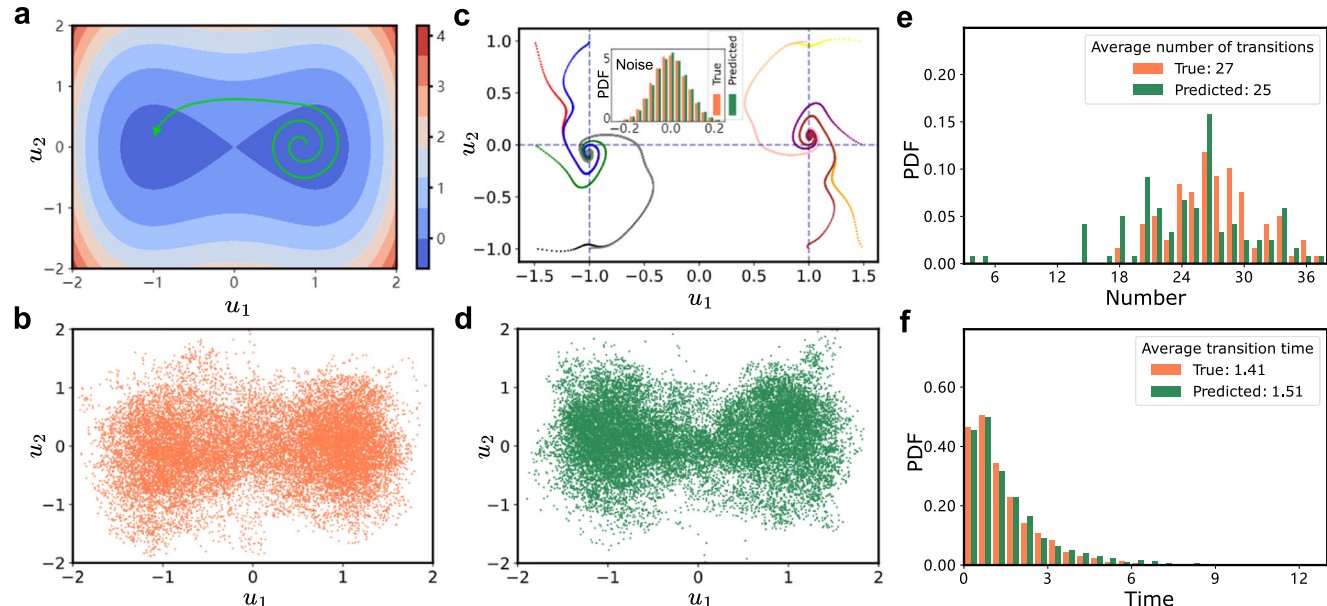

**Fig. 4 | Learning noise-induced transitions in a bistable non-gradient system.**
**a** Schematic of transitions in the 2D bistable non-gradient system. **b** Generated time series from Eqs. (14) and (15) ($a = b = 5$, $c = 0$, $\varepsilon_1 = \varepsilon_2 = 0.3$, $u_1(0) = 0$, $u_2(0) = 2$, $\delta t = 0.002$) with $t = 40$ as the ground truth. **c** The trained slow-scale model transforms ten different start points into ten different slowly time-scale series (color lines), $t \in [40, 80]$, and the noise distribution is separated in the training phase. **d** Result of prediction using the slow-scale model and the noise distribution in **c**. **e** The number of transitions for the 100 replicates simulated in $t \in [40, 80]$ and that from the 100 predicted trajectories matches. **f** Histograms of transition time for the test and predicted data. The transition occurs when the time series crosses the zero point in the $u_1$-direction without returning for $50\delta t$. The transition time is defined as the interval between two consecutive zero crossings.

To evaluate the accuracy of the prediction, we applied the same hyperparameters to conduct 50 predictions as in ref. 40. These predictions were then used alongside trained slow-scale model for 50 times rolling prediction. The average of the 50 predictions outcomes closely approximates the actual time series (Fig. 3d). Furthermore, Fig. 3e shows a near-zero average absolute error between the 50 predictions and the actual time series, indicating high accuracy. These results demonstrate that the present approach requires no assumptions about knowing the deterministic part, underscoring its potential in predicting a single stochastic transition under colored noise.

**A bistable non-gradient system.** We next focus on investigating whether the present method can predict noise-induced transitions in 2D non-gradient systems. We consider a bistable non-gradient system[43]:

$$\dot{u}_1 = -b(-u_1 + u_1^3 + c) - au_2 + \sqrt{2\varepsilon_1 b}\xi_1(t), \quad t \geq 0, \qquad (14)$$

$$\dot{u}_2 = a(-u_1 + u_1^3 + c) - bu_2 + \sqrt{2\varepsilon_2 b}\xi_2(t), \quad t \geq 0, \qquad (15)$$

with Gaussian white noise $\xi_1(t)$ and $\xi_2(t)$. In this system, $b$ is the diffusion coefficient, $a$ represents the strength of the non-detailed balance part, $\varepsilon_1$ and $\varepsilon_2$ are the noise strengths, and $c$ controls the tilt of the potential. The system has noise-induced transitions between the two potential wells under the noise, as illustrated in Fig. 4a. The presence of a non-detailed balance introduces a rotational component to the time series, which complicates the prediction.

In the training phase, we generated a series consisting of $40000\delta t$. The training set is $t \in [0, 40]$, and the predicting set is $t \in [40, 80]$. Figure 4b displays the training set. Following the method in our framework, the deterministic part is reconstructed as shown in Fig. 4c. A proper set of hyperparameters is listed in Table 1 (Example 3). We observe that the generated time series starting from the ten initial points converge to two potential wells, where the time series has

rotational dynamics. In the predicting phase, we perform rolling prediction within $t \in [40, 80]$ (Fig. 4d).

In the evaluation, we predict 100 replicates of the time series, and compare them with 100 replicates simulated from Eqs. (14) and (15). Figure 4e presents histograms of the number of transitions for the 100 predicted and test time series, in $t \in [40, 80]$. Figure 4f presents histograms of transition time for the 100 predicted and test time series, in $t \in [40, 80]$. The results demonstrate that, for the 2D bistable non-gradient system, the present method accurately learns the dynamics and yields precise estimations on the number of transitions and transition time.

**Experimental data of protein folding.** We apply the present method to the protein folding data[6], demonstrating that it can learn the noise-induced transitions of experimental data. The talin protein has five regions with distinct states, and two states (native and unfolded) can be singled out in a short time (native folding dynamics). A short end-to-end length represents the native state, while a longer length represents the unfolded state. This shift of end-to-end length can be considered a noise-induced transition. Figure 5a shows the training data, where transitions occur between two stable states.

In the training phase, with the training set length ($T_{train}$) of 25000 time steps, we obtain the trained slow-scale model and ten different slowly time-scale series with the separated noise distribution (Fig. 5b). The proper hyperparameters are listed in Table 1 (Example 4). In the predicting phase, we employ the trained slow-scale model and the separated noise distribution to do rolling prediction for 100,000 time steps. The first 25000 time steps of prediction are plotted in Fig. 5c, showing the transitions between stable states and the asymmetric dynamics.

In experiments, the available data is often limited, and it is essential to determine the minimum amount of data required. Thus, we reduce the amount of training data to 7500 and 6000 time steps separately. We generate a prediction for 100,000 time steps and then compare it with the test data to evaluate the impact of data length on

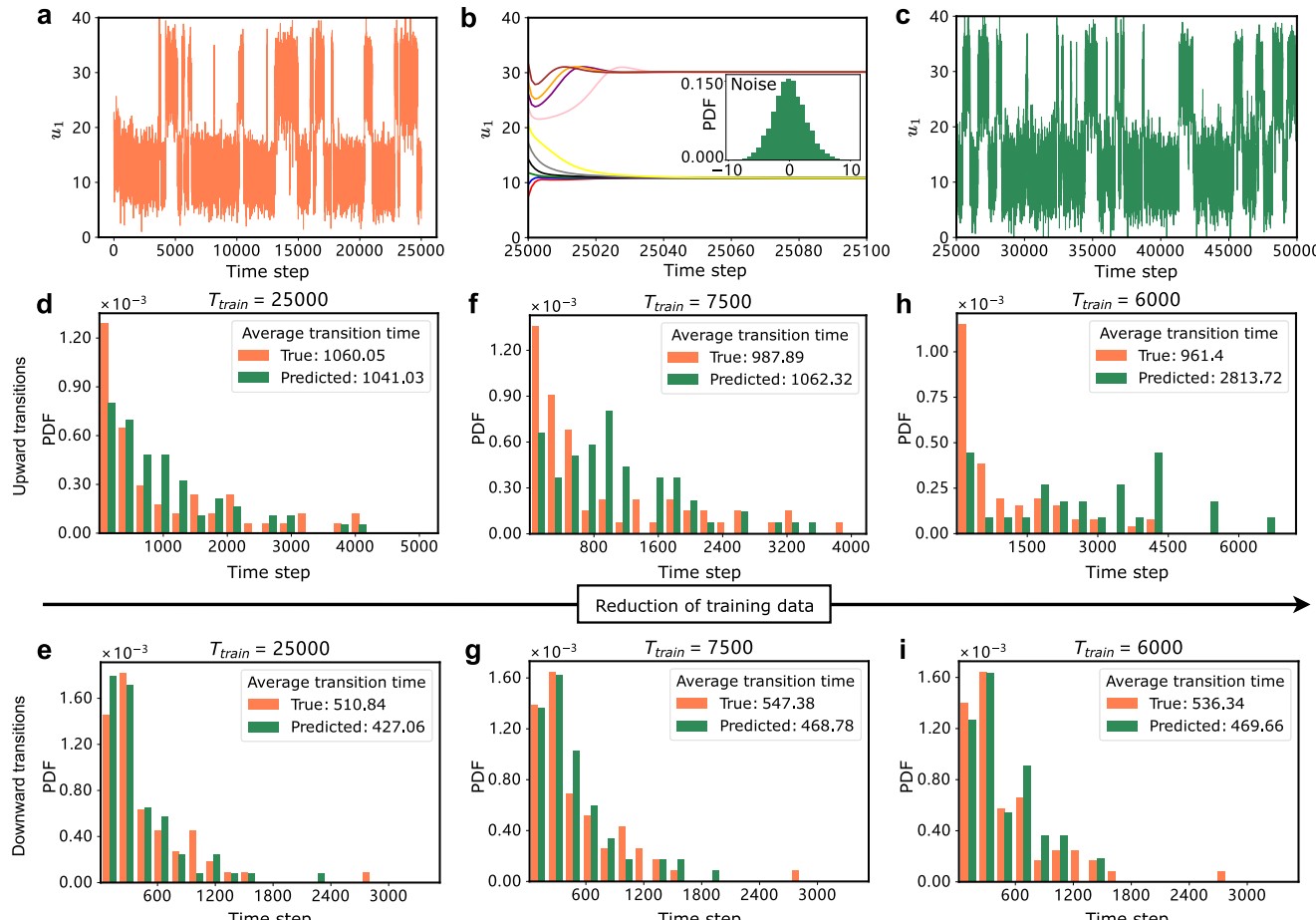

**Fig. 5 | Learning the stochastic transitions from the experimental data of protein folding.** The $u_1$ represents the end-to-end length of the protein. We refer to transitions from around $u_1 = 15$ to around $u_1 = 30$ as upward transitions, and vice versa as downward transitions. The right-pointing arrow: reduction of training data. **a** Time series of the training set ($0 - 25000$ time steps). **b** The trained slow-scale model generates slowly time-scale series (color lines), and the noise distribution is separated out. **c** The prediction during time steps 25000–50,000. **d, e** Histograms of upward and downward transition time for the prediction and the test data, where the length of the training set ($T_{train}$) is 25,000 time steps. Transition time refers to the interval between two consecutive transitions. **f–i** Similar histograms of upward and downward transition time with different lengths of the training sets, $T_{train} = 7500$ for (**f, g**), and $T_{train} = 6000$ for (**h, i**). The present method can still be accurate even when the training length is reduced to $T_{train} = 7500$.

prediction accuracy. The results in Fig. 5d, e and f, g demonstrate that the present method can learn the dynamics of protein folding from the data with around 7500 time steps. Figure 5h, i show a larger error between the predicted and true transition time when $T_{train}$ is equal to 6000 time steps. This suggests that 7500-time steps approximate the minimum data requirement for the present method in this system, allowing the behavior of protein folding to be effectively learned and simulated for more time steps. Additionally, when compared with SINDy-2021 and FORCE learning, our method has higher accuracy (Supplementary Fig. 8). Therefore, the present approach is helpful for streamlining the workload of experimentalists by learning protein folding dynamics from a small dataset.

## Discussion

The choice of hyperparameter $\alpha$ affects the training: the larger $\alpha$ corresponds to the time series with fast time scale, while the smaller $\alpha$ leads to slow time scale[41]. We utilize this characteristic to search for $\alpha$ to match the slow dynamics and separate noise. If a time series is generated from a system with asymmetric potential wells, the two distinct potential wells exhibit different time scales. In this case, we may need to employ two different sets of hyperparameters (Supplementary Figs. 10, 12, 13), where our approach can identify the two-time scales. For colored noise (Example 2), $\alpha$ for the noisy RC is smaller than

that for the deterministic part (Table 1), because using a smaller $\alpha$ leads to a smoother noisy time series and better captures the major trend of colored noise.

The effectiveness of learning slowly time-scale series can also be influenced by other hyperparameters[47]. Although it is challenging to have a universal and systematic strategy for selecting hyperparameters[49], we have proposed a general method to search for the optimal hyperparameters. We find that the power spectral density of the time series can be used to quantify the training performance (Supplementary Fig. 9). The PSD of the predicted stochastic time series closely matches that of the training data when deterministic dynamics are accurately captured. Therefore, a closer match of PSDs indicates a better choice of the hyperparameters. Moreover, we change the values of hyperparameters (Supplementary Figs. 18, 19). The RC is still effective when $\alpha \in [0.15, 0.25]$ and the regularization parameter $\beta$ is not too small. It demonstrates that the method is robust under a range of parameter values.

For experimental data, we have shown the possibility of accurate predictions from a small dataset, as exemplified for the protein folding data[6]. During a short time span, the data samples a local equilibrium involving the native and unfolded conformations. If the measurement time is significantly extended, previously inaccessible regions separated by high-energy barriers may be explored. Consequently, in order to capture a wider variety of protein folding transitions, it may be

necessary to use a longer training set, which needs higher computational cost. Additionally, we observed tilted dynamics in the time series of protein folding. Even so, we can learn both the upward and downward transitions by using only one set of hyperparameters. This suggests that the time scales of upward and downward transitions might not differ significantly. When dealing with a time series generated from very tilted dynamics, we can employ two distinct sets of hyperparameters.

The present method is found effective for various cases of the frequency distributions of the data (Supplementary Fig. 20). For Example 1, the frequency distribution has almost equal intensity over a long range of frequencies due to the white noise. Differently, in Example 2 with colored noise, the frequency distributions of deterministic and noisy signals are mixed with their indistinguishable frequency distributions. To further test the effectiveness of our approach in such cases, we apply it to another case with mixed frequency distributions, where the precise transition time is also accurately predicted. Besides, for the real data of protein folding, the frequency distribution is similar to Example 1, which helps us to better grasp the data characteristics. In general, the frequency distribution of the data can help guide the training, including the choice of the hyperparameters.

In summary, we have provided a general framework for learning noise-induced transitions solely based on data. We have applied the method to examples from stochastic differential equations and experimental data, where the method can accurately learn transition statistics from a small training set. As potential ways of improvements, the Bayesian optimization[50] and simulated annealing[51] can be used to help the search for the hyperparameters. The present approach may be applied to analyze transitions of trajectories between different dynamical phases of spins[52]. The approach can also be generalized to the examples with hidden nodes and hidden links[53], or with other types of noise, where the conditional generative adversarial network[54] may be employed to model the noise. We anticipate that this study can motivate a series of systematic explorations on learning noise-induced phenomena beyond mitigating noisy effect in extracting deterministic dynamics, such as by extending the frameworks of SINDy and FORCE learning.

## Methods

We first describe the previous training process of reservoir computing. We reformulate the loss function to derive the expression for the output matrix $W_{out}$ and discuss the hyperparameters in the present method. The loss function is given as Eq. (5). In detail, we should write the loss function as a sum from all the parameters to do linear regression. Then, the regression becomes simply a sum of vectors:

$$
\begin{aligned}
L &= \sum_{t=1}^{T} [||\mathbf{u}_t - W_{out}\mathbf{r}_t||^2 + \beta ||W_{out}||^2] \\
&= \sum_{t=1}^{T} [(\mathbf{u}_t - W_{out}\mathbf{r}_t)^\top (\mathbf{u}_t - W_{out}\mathbf{r}_t) + \beta ||W_{out}||^2] \\
&= \sum_{t=1}^{T} [(\mathbf{u}_t)^\top \mathbf{u}_t - (W_{out}\mathbf{r}_t)^\top \mathbf{u}_t - (\mathbf{u}_t)^\top W_{out}\mathbf{r}_t + (W_{out}\mathbf{r}_t)^\top W_{out}\mathbf{r}_t + \beta ||W_{out}||^2].
\end{aligned}
\tag{16}
$$

As the loss function is convex (to prove that the zero gradient is indeed the local minimum, one needs to differentiate once more to obtain the Hessian matrix and show that it is positive definite; this is provided by the Gauss-Markov theorem), the optimum solution lies at the zero gradient by

$$
\partial_{W_{out}} L = \sum_{t=1}^{T} [-2(\mathbf{r}_t)^\top \mathbf{u}_t + 2(\mathbf{r}_t)^\top W_{out}\mathbf{r}_t + 2\beta W_{out}] = 0,
\tag{17}
$$

which leads to the regression:

$$
W_{out} = \sum_{t=1}^{T} [(\mathbf{u}_t) \cdot (\mathbf{r}_t)^\top] \cdot [(\mathbf{r}_t) \cdot (\mathbf{r}_t)^\top + \beta]^{-1},
\tag{18}
$$

where we have neglected the notation of the identity matrix and the identify vector. By stacking the vectors of different time points as a vector, it can be rewritten as a compact form Eq. (6). Here, the present method further adjusts $W_{out}$ by tuning hyperparameters to capture the deterministic dynamics and separate noise distribution, thereby enabling us to learn stochastic dynamics.

There are six hyperparameters in the present method. The variable $N$ represents the number of reservoir nodes, which determines the reservoir size. In most cases, performance improves with larger reservoir[55]. However, using large reservoir might lead to overfitting, requiring the application of suitable regularization techniques[47]. The hyperparameter $K_{in}$ represents the scaling factor for the input matrix $W_{in}$. The average degree of the reservoir connection network is denoted by $D$, and we choose the connection matrix $A$ to be sparse[11]. This approach stems from the intuition that decoupling the state variables can result in a richer encoding of the input signal[55]. The spectral radius of the reservoir connection network, denoted as $\rho$, represents a critical characteristic of the dynamics of the reservoir state. Notably, it affects both the nonlinearity of the reservoir and its capacity to encode past inputs in its state[42,55]. The $\alpha$ is the leak parameter, which determines the time scale[41]. The hyperparameter $\beta$ represents the regularization term[47].

## Data availability

The authors declare that the data supporting this study are available within the paper.

## Code availability

A pytorch implementation of the present algorithm can be found in the GitHub repository (https://github.com/Machine-learning-and-complex-systems/NIT-RC).

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

## Acknowledgements

We acknowledge Xingang Wang for helpful discussions. We thank Rafael Tapia-Rojo for sharing the data of protein folding. This work is

supported by Project 12322501, 12105014 of National Natural Science Foundation of China.

## Author contributions

Y.T. had the original idea for this work. Z.Q.L. and Z.F.L. performed the study. Z.Q.L., Z.F.L., Y.T. and Z.R.D. contributed to the preparation of the manuscript.

## Competing interests

The authors declare no competing interests.
