## [Peer Review File · Nature Communications]

Reviewers' comments:

Reviewer #1 (Remarks to the Author):

This paper addresses the problem of predicting noise-induced transitions in multiscale noisy dynamical systems from the time series data of the slow variables.

Previous work (Ref. [29]) uses reservoir computing (RC) and the assumption that the slow deterministic part of dynamics are known. The main novel contribution of this paper is to propose an effective RC based method to remove such an assumption, and therefore is significant for practical applications in science and engineering. The main idea is to make the reservoir multiscale and fine-tune its time scale (the alpha parameter) before training.

Overall the paper reads well (although some sentences could be revised to make the paper more reader-friendly) and the method is simple to understand. Experiments (using both synthetic and real data) are provided to demonstrate the method. However, there are no experiments on data coming from high-dimensional systems (more than 3D, e.g., see the third example in [29]), casting the doubt on the effectiveness of the method on high-dimensional systems.

There are no ablation studies on the effect of the parameter alpha on the results obtained. I believe that the results would be quite sensitive to the value of alpha used. How do we choose alpha in the first place? Do we give an initial guess of say 0.2, see what happen to the result and then fine-tune it by perturbing around this value? It would be useful to provide a comparative study using different values of alpha for each example and consider how they change the quality of the prediction.

Minor comments:

- The value of beta used is very small. What if one increases it? What effects would it have on the results?
- It would be helpful to provide source code that could reproduce the results obtained in the paper

Reviewer #2 (Remarks to the Author):

In "Learning noise-induced transitions by multi-scaling reservoir computing," the authors adapt a reservoir computing (RC) approach for solving the problem of generating data from slow dynamics driven by high-frequency noise. Specifically, the authors use a first-order time-discretized form of the continuous-time RC equations with randomly initialized weights (Eq. 1-3), and tune α ---the effective time scale---alongside other hyperparameters to obtain converging stable states from initial conditions. Then, the authors model the $t+1$ step noise as the residual between the true signal and RNN output, and add samples from this noise-distribution back into the outputs of the RC, which are then used to autonomously evolve the RC. For validation, the authors compute the transition statistics from a variety of stochastic systems including the number of transitions and time to transition in artificially generated and empirical data. The authors look at univariate and bivariate bistable gradient systems driven by stochastic noise (Eq 9), noise generated from a chaotic Lorenz system (Eq 10-13), stochastic noise with a rotation (Eq 14-15), and empirical protein data.

Overall, the authors tackle a challenging and important problem using a multi-scale RC approach. Specifically, deterministic RC approaches have difficulty in stochastic contexts, and the authors provide a nice method for decomposing the problem into slow and fast components to forecast the dynamics with reasonable accuracy. While I found the paper to be insightful and the methods elegant for the audience interested in the use of RCs, most of my comments and concerns revolve around the practical applicability and benchmarking against other non-RC approaches.

Major Comments:

1) Necessity of the RCs for the specific problem: While I agree that these multi-scale RCs are certainly capable of learning noise-induced transitions, the RC with manually-tuned α is simply a substitute for a more principled and classical optimization on an ansatz of the functional form of the slow dynamics. Said another way, the RC is essentially a nonlinear basis generating machine, where the hyperparameters (such as α) tune the time-scale of the represented basis [1]. As such, the primary machine learning portion of the paper of generating a good basis via tuning α may in theory be performed through methods such as Sindy [2], which may also yield interpretable dynamics through the learned coefficients. I believe the paper would be significantly strengthened if the authors can demonstrate that an earnest attempt at estimating the slow time-scale dynamics via Sindy yields either less accurate or incorrectly identified terms. Then the practical advantage of the RC approach would be superior estimation of transition statistics.

2) Manual tuning of alpha: Additionally, this alpha parameter is manually tuned in the paper, which seems to me no different than manually generating and tuning an explicit basis. The fact that an RC is capable of generating a basis on which a linear output is trained seems obvious to me, and the choice of tuning via RC vs an analytic approach such as Sindy seems more of a stylistic preference. Perhaps the situation would be improved if the authors automated the meta-learning of hyperparameters via some additional cost? Perhaps the authors could parameterize the noise distribution under a general family of distributions, and jointly learn alpha, Wout, and the distribution parameters?

3) Assumed separability of high- and low-frequency signals: There are a lot of assumptions that are made about the data. While some data really are comprised of low-frequency transitions and high-frequency noise that are easily separated, I believe that this RC approach will have substantial difficulty with signals where the noise plagues the resonant frequency of the dynamical system of interest, which is the most challenging subset of problems that the authors claim to address. Can the authors demonstrate performance as the frequency band of the noise signal bleeds into the frequency range of the natural dynamics? The goal would not be to get the RC to succeed at an impossible task (although it would be amazing if it did!) but rather to provide the reader with an expectation of whether or not their data is suitable for this method.

4) Some existing and related (though by no means identical) approaches for modeling sequences: other methods for handling sequence generation for noisy data using general recurrent neural networks (RNNs) perform similar (though not identical) types of tasks on noisy data [3]. Additionally, I would like to see some benchmarking of the authors' multiscale RC method to other simpler methods (can even be very naive ones such as zero-phase low-pass filtering the data and performing Sindy). Such a comparison would accentuate the contribution of this RC approach.

Minor Comments:

1) Methodological clarity: there were a few points of confusion for me.

a) Eq. 1 and 2 suggest the use of a discrete-time RC, but Eq. 3 and the fact that all subsequent models were continuous-time leads me to believe that the authors instead used the continuous-time RC in Eq. 3. This suspicion is accentuated given how large alpha is in Table 1. Unless I've misunderstood, the authors should make clear whether they are using a true discrete-time RC, or the continuous-time formulation with smaller step-size integration.

b) It took a while for me to understand the methods involving the second RC in Figure 3. It would be worth adding more methodological detail about this second RC, along with some discussion about how that second deterministic RC can only model colored noise because the noise source itself is a deterministic dynamical system.

2) More discussion about the protein data: while I found the previous results and methods to be elegant, I thought the prediction of protein transitions was the most impactful and meaningful result. I recommend that the authors provide more detail about what are the stable states, what is the protein, and the biological context of the data.

3) Spelling and grammar: throughout, I found multiple spelling and grammar mistakes. I assume the editorial team will manage these, but I thought it important to bring them up just in case.

References:

[1] Gauthier, Daniel J., et al. "Next generation reservoir computing." *Nature communications* 12.1 (2021): 5564.

[2] Brunton, Steven L., Joshua L. Proctor, and J. Nathan Kutz. "Discovering governing equations from data by sparse identification of nonlinear dynamical systems." *Proceedings of the national academy of sciences* 113.15 (2016): 3932-3937.

[3] Sussillo, David, and Larry F. Abbott. "Generating coherent patterns of activity from chaotic neural networks." *Neuron* 63.4 (2009): 544-557.

Reviewer #3 (Remarks to the Author):

In this paper, the authors demonstrated that reservoir computing can be used to learn noise-induced transitions. In particular, through hyperparameter optimization, their trained reservoir computer model can generate accurate statistics of the transition time. During training, one of the hyperparameters (α) is tuned to match the slow time scale dynamics, enabling the fast time-scale variations to be separated as noise and saved. During testing, the slow scale dynamics are predicted and the saved noises are added back to the prediction. The authors tested the machine-learning model using a double-well potential mechanical system, the asymmetrical version of this system, and experimental data of protein folding.

This paper lacks originality and appears to be conceptually flawed. It is not suitable for publication in Nature Communications.

1. Separating noise from the slow dynamics during training and then adding it back during the testing make no physical sense. The training data are typically collected prior to deploying the machine-learning algorithm for testing, i.e., training and testing occur in different time periods. How can the noise be the same during two different time periods?

2. If the goal is to separate noise from the deterministic dynamics, it is not necessary to use reservoir computing. If the frequency bands of the deterministic evolution and of the noise are well separated, a simple filtering can do the job. If that is not the case, there is large literature on noise reduction and mitigation in nonlinear dynamical systems.

3. Tuning hyperparameters using certain optimization procedures has been a well established practice in research on reservoir computing. The authors regarded this practice as innovative, which is inappropriate.

4. Reservoir computing has been demonstrated to be capable of accurate prediction of the critical transitions and bifurcations in nonlinear dynamical systems [e.g., Phys. Rev. Research 3, 013090 (2021); Chaos 33, 033111 (2023)]. Not only the transition point but also the statistical distribution of the transient time after the transition can be predicted. Noise-induced transitions are one type of transitions in nonlinear dynamical systems. It is justifiably anticipated that reservoir computing should be able to be trained to predict noise-induced transitions. In addition, the beneficial role of noise in enhancing reservoir computers' ability to predict chaotic system has also been known [Phys. Rev. Research 5, 033127 (2023)].

Responses to Reviewers - NCOMMS-23-43211

We thank all reviewers for their careful reading and constructive comments, which lead to a better manuscript. We are glad that reviewers 1 and 2 are generally positive: Reviewer 1 found that the manuscript has a significant contribution and reads well; Reviewer 2 commented that the manuscript “tackle a challenging and important problem”, and is insightful and elegant. Reviewer 3 had concerns on the extent of originality, and accordingly we have carefully revised the manuscript to eliminate possible misunderstanding, such that it becomes clearer that a significant progress has been made over the previous related approaches.

An overview of the changes is given below, followed by point-to-point responses to reviewers’ comments. The original reports are typed in orange, our response in black, and the revised text in blue. Please note that the numbers of equations and figures refer to the revised manuscript. The quoted references are listed at the end of the response.

List of main modifications in the revised manuscript

1. Abstract:

- Revised the presentation to clarify the progress of the present method compared with previous methods.

2. Introduction:

- Summarized our attempts of applying the conventional methods, including sparse identification of nonlinear dynamics (SINDy), Kalman filter, and recurrent neural network (RNN), to learn noise-induced transitions, and their ineffectiveness in such problems.

3. Figures:

- Modified FIG.1 for better presentation of the protocol: revised the “Train” to “After trained”; added dashed lines to denote the procedure of rolling prediction.

4. Results:

- Added the results of applying the related approaches to learn stochastic transitions.
- Elaborated the motivation for introducing a second RC for the case with colored noise.
- Demonstrated in the Example 2 that the present method can accurately predict a single transition even when data frequencies are mixed.
- Provided additional details about the experimental data of protein folding.
- To analyze the noise type (colored or white) of experimental data, we utilized Fourier transform to get the frequency distribution of the data.

5. Discussion:

- Demonstrated that the present method can learn noise-induced transitions when frequencies of data are mixed.
 - Discussed the robustness of present method under a range of hyperparameters, as exemplified for α and β .
6. Added references for conventional methods, such as SINDy [1] and RNN [2]. We also cited code libraries, including the SINDy library Pysindy [3], the RNN library [4], and the filter library [5, 6].
7. Supplementary Information:
- Introduced Supplementary FIG. 1 to compare various existing approaches, including SINDy, RNN, and Kalman filter.
 - Provided more details for the comparison with the previous approaches (Supplementary FIGs. 2, 3, 4).
 - Added Supplementary FIG. 5 about the frequency analysis of the time series (Example 2 of the main text) by Fourier transform, demonstrating that the present method can predict single transitions when frequency distributions of deterministic and noisy signals are mixed.
 - Showcased an example of a bistable system with the high-dimensional Lorenz-96 system noise (Supplementary FIG. 13).
 - Added Supplementary FIGs. 14-15 to demonstrate robustness of the method under the variation of the hyperparameters α and β .

Response to Reviewer 1

This paper addresses the problem of predicting noise-induced transitions in multiscale noisy dynamical systems from the time series data of the slow variables.

Previous work (Ref. [29]) uses reservoir computing (RC) and the assumption that the slow deterministic part of dynamics are known. The main novel contribution of this paper is to propose an effective RC based method to remove such an assumption, and therefore is significant for practical applications in science and engineering. The main idea is to make the reservoir multiscale and fine-tune its time scale (the α parameter) before training.

Response: We thank the reviewer for the careful reading on our manuscript. We are also grateful for the reviewer’s pithy summary of the major results and the encouraging comment of finding the contribution of the manuscript significant.

Overall the paper reads well (although some sentences could be revised to make the paper more reader-friendly) and the method is simple to understand. Experiments (using both synthetic and real data) are provided to demonstrate the method. However, there are no experiments on data coming from high-dimensional systems (more than 3D, e.g., see the third example in [7]), casting the doubt on the effectiveness of the method on high-dimensional systems.

Response: We are glad that the reviewer found the paper to be well-written, and we have further improved the presentation of the manuscript. The reviewers also suggested testing the present method in high-dimensional systems more than 3D, such as the third example in [7]. Following the reviewer’s suggestion, we now have applied the method to a bistable system subject to noise from a high-dimensional Lorenz-96 system. The result shows that the present method also accurately captures the stochastic transition with high-dimensional noise (Supplementary FIG. 13).

We would like to mention that although this example is a high-dimensional system, the observed variable is still in one dimension, as the other variables are about the noise. As in the Fig. 4 of the example [7], they mainly predicted the stochastic transition for the one-dimensional observed time series. Beyond such a case, the present method is also applicable to the case with higher-dimensional observed time series. For example, we have predicted the two-dimensional time series without detailed balance, supporting that the method is potentially applicable to higher dimensional observed data.

There are no ablation studies on the effect of the parameter α on the results obtained. I believe that the results would be quite sensitive to the value of α used. How do we choose α in the first place? Do we give an initial guess of say 0.2, see what happen to the result and then fine-tune it by perturbing around this value? It would be useful to provide a comparative study using different values of α for each example and consider how they change the quality of the prediction.

Response: The reviewer raised an important question. Yes, the present result depends on

the choice of hyperparameters such as α . Now we have elaborated the procedure for choosing the proper α . Specifically, one can start with the values of $\alpha \in [0, 1]$, and then fine-tune it by trial and error around the initial value. For instance, in the Example 1 of the main text, we have compared the results under perturbations of α around the initial value, which enables us to explore a better choice of α . Remarkably, we find that the present method can provide robust prediction under a range of this hyperparameter, $\alpha \in [0.15, 0.25]$ as shown in Response FIG. 1 and Response FIG. 2.

Response FIG. 1. **The robustness of the results in the training phase under the hyperparameters α and β of the RC.** In the Example 1, the hyperparameters $\alpha = 0.2, \beta = 1 \times 10^{-8}$. (a) The trained slow-scale model transforms ten different initial points into ten different slowly time-scale series with $\alpha = 0.15, \beta = 1 \times 10^{-8}$. It presents an accurate prediction on the deterministic part of dynamics. (b) Result with $\alpha = 0.2, \beta = 1 \times 10^{-8}$. (c) Result with $\alpha = 0.25, \beta = 1 \times 10^{-8}$. (d-f) The results are the same as in (a-c), with $\beta = 1 \times 10^{-7}$. (g-i) Results same as (a-c), while $\beta = 1 \times 10^{-6}$.

Minor comments:

- The value of beta used is very small. What if one increases it? What effects would it have on the results?

Response:

Response FIG. 2. **The robustness of the results in the evaluation under the hyperparameters α and β of the RC.** In the Example 1 of the main text, the hyperparameters $\alpha = 0.2, \beta = 1 \times 10^{-8}$. The length of the predicting set is $10000\delta t$, while $\delta t = 0.01$. PDF: probability density function. (a) Histograms of the transition time for the test and predicted data with $\alpha = 0.15, \beta = 1 \times 10^{-8}$. The results demonstrate the ability to capture stochastic transitions under the values of α and β . (b) Results with $\alpha = 0.2, \beta = 1 \times 10^{-8}$. (c) Results with $\alpha = 0.25, \beta = 1 \times 10^{-8}$. (d-f) The results are the same as in (a-c), with $\beta = 1 \times 10^{-7}$. (g-i) Results same as (a-c), while $\beta = 1 \times 10^{-6}$.

Thanks for the question. We have increased β from 10^{-8} to 10^{-7} and even 10^{-6} in Response FIG. 1 (Supplementary FIG. 14), to show its effect on the result. The learned deterministic dynamics converge to the correct bistable fixed points, and the predicted transition statistics are also accurate (Response FIG. 2). Thus, the result is robust over a range of β values. We further remark that β should not be chosen too small, because it is a regularization parameter and a too small value does not fulfill the role of regularization. Thus, in the literature, β was typically chosen in the range of $[10^{-10}, 10^{-2}]$ ([8, 9, 10]) and is system-dependent. Here, our choices on β are within its typical range and give consistent results.

- It would be helpful to provide source code that could reproduce the results obtained in the paper.

Response: Certainly. Previously, we have provided the code through the journal-suggested Figshare platform. Now, we have also included the source code as an additional review material.

Finally, we thank the reviewer again for the careful reading and constructive suggestions.

Response to Reviewer 2

In “Learning noise-induced transitions by multi-scaling reservoir computing”, the authors adapt a reservoir computing (RC) approach for solving the problem of generating data from slow dynamics driven by high-frequency noise. Specifically, the authors use a first-order time-discretized form of the continuous-time RC equations with randomly initialized weights (Eqs. 1-3), and tune α —the effective time scale—alongside other hyperparameters to obtain converging stable states from initial conditions. Then, the authors model the $t + 1$ step noise as the residual between the true signal and RNN output, and add samples from this noise-distribution back into the outputs of the RC, which are then used to autonomously evolve the RC. For validation, the authors compute the transition statistics from a variety of stochastic systems including the number of transitions and time to transition in artificially generated and empirical data. The authors look at univariate and bivariate bistable gradient systems driven by stochastic noise (Eq. 9), noise generated from a chaotic Lorenz system (Eqs. 10-13), stochastic noise with a rotation (Eqs. 14-15), and empirical protein data.

Response: We thank the reviewer for the careful reading of our manuscript. We are also grateful for the reviewers’ pithy summary of the major results.

Overall, the authors tackle a challenging and important problem using a multi-scale RC approach. Specifically, deterministic RC approaches have difficulty in stochastic contexts, and the authors provide a nice method for decomposing the problem into slow and fast components to forecast the dynamics with reasonable accuracy. While I found the paper to be insightful and the methods elegant for the audience interested in the use of RCs, most of my comments and concerns revolve around the practical applicability and benchmarking against other non-RC approaches.

Response: We are glad that the reviewer recognized that the problem is challenging and important. We appreciate the reviewer’s positive assessment that the manuscript is insightful and the methods are elegant. Below, we have carefully addressed the reviewer’s questions and revised the manuscript accordingly.

Major Comments:

1) Necessity of the RCs for the specific problem: While I agree that these multi-scale RCs are certainly capable of learning noise-induced transitions, the RC with manually-tuned α is simply a substitute for a more principled and classical optimization on an ansatz of the functional form of the slow dynamics. Said another way, the RC is essentially a nonlinear basis generating machine, where the hyperparameters (such as α) tune the time-scale of the represented basis [9]. As such, the primary machine learning portion of the paper of generating a good basis via tuning α may in theory be performed through methods such as Sindy [1], which may also yield interpretable dynamics through the learned coefficients. I believe the paper would be significantly strengthened if the authors can demonstrate that an earnest attempt at estimating the slow time-scale dynamics via Sindy yields either less accurate or incorrectly identified terms. Then the practical advantage of the RC approach

would be superior estimation of transition statistics.

Response FIG. 3. **The previous related approaches are unable to capture stochastic transitions.** (a) Upper: a piece of training data for SINDy-2021 and RNN. Lower: filtered data from upper for SINDy-2016 to learn the filtered time series. (b) Results of the predictions. Upper: predicted data from SINDy-2021 (pale purple) and RNN (deep purple). Lower: the prediction by SINDy-2016 on the filtered data (cyan).

Response: Thanks for the great suggestion. We now have applied SINDy in [3, 11] to the problem of learning noise-induced transitions. Specifically, we have tested the two different versions of SINDy for the Example 1 of the main text, including the version at the year 2016 designed for the deterministic dynamics (SINDy-2016) and that at the year 2021 for data with noise (SINDy-2021). In the latter, they tended to mitigate the effect of noise on learning deterministic dynamics and separate the noise distribution.

In Response FIG. 3, we have applied filters to the data with noise and employed the filtered data for training SINDy-2016, however, the predicted time series converge falsely. Besides, SINDy-2021 does not accurately identify the deterministic equation (Response Table 1). After adding the noise back to the deterministic part, SINDy-2021 does not predict stochastic transitions, or learn the actual transition time (Supplementary FIG. 2), even for a one-dimensional bistable system. The ineffectiveness of SINDy-2021 was found under the choices

of a set of hyperparameter values (Response Table 1). Instead, the present method can fulfill this task (Supplementary FIG. 2).

Response Table 1. Identifications of SINDy from data with noise.

SINDy types	Set	threshold	Actual equation	Function
SINDy-2016	1	0.01	$\dot{u}_1 = -5(-u_1 + u_1^3)$	$\dot{u}_1 = -0.0191 - 0.013u_1^3$
SINDy-2016	2	0.005	$\dot{u}_1 = -5(-u_1 + u_1^3)$	$\dot{u}_1 = -0.0111 - 0.01u_1^2 - 0.013u_1^3$
SINDy-2021	3	0.005	$\dot{u}_1 = -5(-u_1 + u_1^3)$	$\dot{u}_1 = -0.0157u_1^2 - 0.0184u_1^3$
SINDy-2021	4	0.001	$\dot{u}_1 = -5(-u_1 + u_1^3)$	$\dot{u}_1 = -0.0039u_1 - 0.0158u_1^2 - 0.0133u_1^3$

2) Manual tuning of α : Additionally, this α parameter is manually tuned in the paper, which seems to me no different than manually generating and tuning an explicit basis. The fact that an RC is capable of generating a basis on which a linear output is trained seems obvious to me, and the choice of tuning via RC vs an analytic approach such as Sindy seems more of a stylistic preference. Perhaps the situation would be improved if the authors automated the meta-learning of hyperparameters via some additional cost? Perhaps the authors could parameterize the noise distribution under a general family of distributions, and jointly learn α , W_{out} , and the distribution parameters?

Response: The reviewers mentioned an insightful understanding on finding the proper hyperparameters. First, we would like to note that manual tuning is widely used in training reservoir computing for many dynamical systems and performs well [12, 9]. Second, now we have applied SINDy to the Example 1 of the main text. We find that neither SINDy-2016 nor SINDy-2021 accurately identify the deterministic equation and capture stochastic transitions (Supplementary FIG. 2), indicating that SINDy may not be suitable for this type of problems.

As the reviewer suggested, these hyperparameters may also be set as meta-learning parameters with parameterizing the noise distribution, such as by constructing a loss function for optimizing hyperparameters [11]. We have considered this proposal and found that by directly analyzing the data by its frequency distribution, we already have a clue on choosing the proper α , which works well for separating the deterministic dynamics and learning the noise distribution in the examples (see the response to the next point). Thus, we would like to leave the point of parameterizing the noise distribution as an extension in the future work.

3) Assumed separability of high- and low-frequency signals: There are a lot of assumptions that are made about the data. While some data really are comprised of low-frequency transitions and high-frequency noise that are easily separated, I believe that this RC approach will have substantial difficulty with signals where the noise plagues the resonant frequency of the dynamical system of interest, which is the most challenging subset of problems that the authors claim to address. Can the authors demonstrate performance as the frequency band of the noise signal bleeds into the frequency range of the natural dynamics? The goal would not be to get the RC to succeed at an impossible task (although it would be amazing if it

did!) but rather to provide the reader with an expectation of whether or not their data is suitable for this method.

Response: Following the reviewer’s suggestion, we now have studied the case where the frequencies of the noise signal are mixed into the frequency distribution of the deterministic dynamics. Specifically, we have applied the Fourier transform to the data. For the Example 2 of the main text, the frequencies of deterministic dynamics and noise are already mixed (Response FIG. 4(b)). It demonstrates that the presented method can perform well even in such a difficult case.

To further demonstrate the effectiveness of the method in handling mixed frequency distributions, we adjusted the noise parameters to generate another example. The results in Response FIG. 4(c-e) (Supplementary FIG. 5) show that the present method can also accurately predict a single transition for this example. Therefore, the current method performs effectively when data frequencies are mixed, as exemplified in the bistable system with Lorenz noise.

Response FIG. 4. **Results of present method when the frequencies of the time series are mixed.** (a) Upper: generated time series from Eq. (9) of the main text (FIG. 2(b)). Lower: fast Fourier transform (FFT) result. When white noise causes the transitions, the FFT results have almost equal intensity at different frequencies except for deterministic dynamics. Amp: amplitude. (b) Upper: generated time series from Eqs. (10)-(13) of the main text (FIG. 3(b)). Lower: FFT result. The noise plagues the resonant frequency of the dynamical system of the 1D bistable system. (c) Upper: generated time series (Eqs. (10)-(13)) by adjusting the parameters of the noise ($\epsilon = 1.4, \psi = 0.13$). Lower: FFT result. The frequencies are also mixed. (d) 50 predictions of (c) $u_1(t)$ (fainter lines) with the same hyperparameters. The average prediction (thick green) matches the test data (coral). (e) Absolute error of the predictions. (f) Upper: experimental data (FIG. 5(a) of the main text). Lower: FFT result, which presents that white noise influences the dynamics.

4) Some existing and related (though by no means identical) approaches for modeling sequences: other methods for handling sequence generation for noisy data using general recurrent neural networks (RNNs) perform similar (though not identical) types of tasks on noisy data [2]. Additionally, I would like to see some benchmarking of the authors' multiscale RC method to other simpler methods (can even be very naive ones such as zero-phase low-pass filtering the data and performing Sindy). Such a comparison would accentuate the contribution of this RC approach.

Response: Thanks for the suggestion. We now have compared the RNN approach and the low-pass filtering approach. The RNN approach requires more computational cost (Supplementary Table VIII) and does not accurately learn noise-induced transitions (Supplementary FIG. 3). By using the Kalman filter [5] and Savitzky-Golay filter [6] on the training data, we then input the data to SINDy-2016 and RC as shown in Response FIG. 3 and Supplementary FIG. 4. The results demonstrate that the filter and RNN methods, within our attempts under the sets of hyperparameters, do not give accurate predictions for this problem.

We have also compared with SINDy-2021, as demonstrated in the above response. The results show that these previous methods are not directly applicable to this problem even in a 1D bistable system, and need more computational time compared with the present method (Supplementary Table VIII).

Minor Comments:

1) Methodological clarity: there were a few points of confusion for me.

a) Eq. 1 and 2 suggest the use of a discrete-time RC, but Eq. 3 and the fact that all subsequent models were continuous-time leads me to believe that the authors instead used the continuous-time RC in Eq. 3. This suspicion is accentuated given how large α is in Table 1. Unless I've misunderstood, the authors should make clear whether they are using a true discrete-time RC, or the continuous-time formulation with smaller step-size integration.

Response: We apologize for the confusion. We presented the continuous-time RC in Eq. (3) only for demonstrating the role of α , which better shows that α corresponds to the time scale in the RC dynamics. Now we have mentioned this point before Eq. (3). When implementing the RC for the learning task, we employed the discrete-time type of the RC, with specific time step sizes for given examples. For example, we used $\delta t = 0.01$ in Example 2 of the main text, which is the same as Example 1 in [7].

b) It took a while for me to understand the methods involving the second RC in Figure 3. It would be worth adding more methodological detail about this second RC, along with some discussion about how that second deterministic RC can only model colored noise because the noise source itself is a deterministic dynamical system.

Response: Thanks for the great suggestion. We now have added more demonstrations about the second RC in FIG. 3 of the main text. The reason for introducing the second RC is that for colored noise, one needs to learn the time evolution of the separated noise. Under this

case, we introduced the second RC, which learns the time evolution of noise. In Fig. 3 of the main text, the colored noise is given by another deterministic system. Since RC is capable of learning dynamical systems, the second RC successfully learns the time evolution of colored noise. In the main text, it now reads: “We need to learn the time evolution of the separated noise. Since RC is good at learning dynamical systems, we employ a second RC (first RC for deterministic part) to learn the noise series for predicting a single transition.”

We also mention that we follow exactly the same setting of the example in [7], to demonstrate the validity of the present method for the case with colored noise. Notably, the previous work [7] relies on an assumption that the slow deterministic part of dynamics are known. Here, we do not make such an assumption: we use the first RC to learn the deterministic dynamics, separate the noise, and then use the second RC to further track the colored noise. Combining both RCs gives a full prediction on the original dynamics.

2) More discussion about the protein data: while I found the previous results and methods to be elegant, I thought the prediction of protein transitions was the most impactful and meaningful result. I recommend that the authors provide more detail about what are the stable states, what is the protein, and the biological context of the data.

Response: We agree with the reviewer and now provided more details about the example of protein folding. We have presented the biological context of the data for the protein conformation change: “The talin protein has five regions of distinct states, and two states (native and unfolded) can be singled out in a short time (native folding dynamics). A short end-to-end length represents the native state, while a longer length represents the unfolded state. The study of the transitions between these two states contributes to the understanding of cellular dynamics.” With this description, it is more evident that predicting stochastic transitions among states is important for such systems.

3) Spelling and grammar: throughout, I found multiple spelling and grammar mistakes. I assume the editorial team will manage these, but I thought it important to bring them up just in case.

Response: Thanks for pointing this out. We now have provided a clearer presentation and have carefully revised grammar of the manuscript. For example, we revise “Separate the noise distribution in the training phase” to “and the noise distribution is separated in the training phase”. Finally, we thank the reviewer again for the constructive comments and thoughtful suggestions.

Response to Reviewer 3

In this paper, the authors demonstrated that reservoir computing can be used to learn noise-induced transitions. In particular, through hyperparameter optimization, their trained reservoir computer model can generate accurate statistics of the transition time. During training, one of the hyperparameters (α) is tuned to match the slow time scale dynamics, enabling the fast time-scale variations to be separated as noise and saved. During testing, the slow scale dynamics are predicted and the saved noises are added back to the prediction. The authors tested the machine-learning model using a double-well potential mechanical system, the asymmetrical version of this system, and experimental data of protein folding.

Response: We thank the reviewer for the careful reading on our manuscript.

This paper lacks originality and appears to be conceptually flawed. It is not suitable for publication in Nature Communications.

Response: We respectfully disagree on the comment, based on the response to each of the reviewer’s comments below. In agreement with our view, the first two reviewers recognized the new aspects of the manuscript: the first reviewer found the contribution of the manuscript novel, and the second reviewer found the manuscript insightful and method elegant.

1. Separating noise from the slow dynamics during training and then adding it back during the testing make no physical sense. The training data are typically collected prior to deploying the machine-learning algorithm for testing, i.e., training and testing occur in different time periods. How can the noise be the same during two different time periods?

Response: We are afraid that we could not agree with the reviewer on this point. In the field of time series forecast, a fundamental assumption is that what are learnt from the training data can be extrapolated to the testing time period. Following this basic logic, once the setup of the dynamical process does not change dramatically, the noise statistics from the training data can be used for the test data, and this strategy has been widely used in different areas. For example, there are a class of methods to learn stochastic differential equations (SDE) with noise from data [13, 14]. These studies assumed that the noisy distribution is same in the training and predicting periods. Besides, in other research areas such as quantum error correction, many noise models have been developed in order to capture the noise distribution from the training data and mitigate the noise in the new dataset [15]. Based on these studies, learning the noise statistics from the training data for the test data is a reasonable protocol.

2. If the goal is to separate noise from the deterministic dynamics, it is not necessary to use reservoir computing. If the frequency bands of the deterministic evolution and of the noise are well separated, a simple filtering can do the job. If that is not the case, there is large literature on noise reduction and mitigation in nonlinear dynamical systems.

Response: We thank the reviewer for the comment. First, our goal is not to separate noise from deterministic dynamics, but rather to predict the stochastic transitions. For this task, we find that the previous related approaches are ineffective, even for a one-dimensional

bistable system with white noise (Response FIG. 3). Specifically, we have utilized the Kalman filter [5] and the Savitzky-Golay filter [6] to separate the deterministic part, and trained SINDy-2016 [3] or RC to learn the filtered data (Supplementary FIG. 4). These approaches do not capture stochastic transitions even after filtering the noise (Response FIG. 3). We have also applied the noise mitigation method, such as in SINDy [1, 11], which still does not accurately learn the stochastic transitions (Response FIG. 5).

Thus, new methods are required, and here we leverage the reservoir computing for the task. Since the reservoir computing was designed for the deterministic dynamics, we need to develop a procedure of separating the noise. We found that this procedure successfully separate the noise from the deterministic dynamics and predict stochastic transitions. Now, we have further analyzed the frequency distributions of the deterministic and noisy signals: their frequency distributions are mixed as shown in Response FIG. 4 (Supplementary FIG. 5). For such more challenging cases, while the previous filtering or noise mitigation methods are unable to capture stochastic transitions, the present method can fulfill the task.

Response FIG. 5. **Applying conventional approaches for predicting time series preprocessed by noise filtering.** The training data is generated from Eq. (9) of the main text. (a) The actual time series (coral line) and the filtered data (blue line) preprocessed by the Kalman filter (measurement noise covariance matrix $Q = 0.05$). The filtered series is used as the training data of SINDy-2016 and RC. (b) Prediction from SINDy-2016 (purple line) and RC (green line) is significantly different from the filtered data. (c-d) Results are the same as (a-b), and $Q = 0.0005$. (e) Result of preprocessing the time series by the Savitzky-Golay filter (window length $W = 10$, polynomial order $P = 2$). (f) Results of the predicted data (SINDy-2016 and RC) also exhibit differences. (g-h) Same as (e-f), with $W = 50$ and $P = 2$.

3. Tuning hyperparameters using certain optimization procedures has been a well established practice in research on reservoir computing. The authors regarded this practice as innovative, which is inappropriate.

Response: We would like to clarify that we did not regard the procedure of tuning the hyperparameter as innovative. Instead, what we regard as new is that we found a protocol for predicting stochastic transitions by leveraging reservoir computing. Specifically, the protocol of tuning the hyperparameter α for controlling the time scale and separating the noise distribution has not been reported in the previous literature. To our best knowledge, this is the first approach based on reservoir computing to deal with stochastic transitions without assuming prior known information on the dynamics.

4. Reservoir computing has been demonstrated to be capable of accurate prediction of the critical transitions and bifurcations in nonlinear dynamical systems [e.g., Phys. Rev. Research 3, 013090 (2021); Chaos 33, 033111 (2023)]. Not only the transition point but also the statistical distribution of the transient time after the transition can be predicted. Noise-induced transitions are one type of transitions in nonlinear dynamical systems. It is justifiably anticipated that reservoir computing should be able to be trained to predict noise-induced transitions. In addition, the beneficial role of noise in enhancing reservoir computers' ability to predict chaotic system has also been known [Phys. Rev. Research 5, 033127 (2023)].

Response: We thank the reviewer for bringing the relevant literature. We have carefully checked them and found that indeed they studied a different class of problems than stochastic transitions. Specifically, the paper [16] mainly investigated the critical transitions due to parameter drift, which is different from stochastic transitions induced by noise. The other paper [17] utilized reservoir computing as a digital twin for nonlinear dynamics such as Lorenz-96 system to forecast bifurcations and provide early warning signals, but it did not predict noise-induced transitions. In [18], the authors injected noise to the training data to induce stochastic resonance for increasing prediction accuracy, which also differs from the problem of predicting stochastic transitions.

Without an existing study on this class of problems, we could not agree with the reviewer on that one could “anticipate that reservoir computing should be able to be trained to predict noise-induced transitions”. In fact, the ineffectiveness of the conventional approaches, such as SINDy and RNN (Response FIG. 3, Supplementary FIGs. 1-4), show that the problem is highly non-trivial. Finally, we thank the reviewer again for the reading and constructive comments.

References

- [1] Steven L Brunton, Joshua L Proctor, and J Nathan Kutz. Discovering governing equations from data by sparse identification of nonlinear dynamical systems. Proc. Natl. Acad. Sci., 113(15):3932–3937, 2016.
- [2] David Sussillo and Larry F Abbott. Generating coherent patterns of activity from chaotic neural networks. Neuron, 63(4):544–557, 2009.
- [3] Alan A Kaptanoglu, Brian M de Silva, Urban Fasel, Kadierdan Kaheman, Andy J Goldschmidt, Jared L Callahan, Charles B Delahunt, Zachary G Nicolaou, Kathleen Champion, Jean-Christophe Loiseau, et al. Pysindy: A comprehensive python package for robust sparse system identification. arXiv preprint arXiv:2111.08481, 2021.
- [4] Navin Kumar Manaswi. Understanding and Working with Keras, pages 31–43. Apress, Berkeley, CA, 2018.
- [5] Roger R Labbe. Filterpy documentation, 2018.
- [6] Pauli Virtanen, Ralf Gommers, Travis E Oliphant, Matt Haberland, Tyler Reddy, David Cournapeau, Evgeni Burovski, Pearu Peterson, Warren Weckesser, Jonathan Bright, et al. Scipy 1.0: fundamental algorithms for scientific computing in python. Nat. Methods, 17(3):261–272, 2020.
- [7] Soon Hoe Lim, Ludovico Theo Giorgini, Woosok Moon, and J. S. Wettlaufer. Predicting critical transitions in multiscale dynamical systems using reservoir computing. Chaos, 30(12):123126, 2020.
- [8] Jan Yperman and Thijs Becker. Bayesian optimization of hyper-parameters in reservoir computing. arXiv preprint arXiv:1611.05193, 2016.
- [9] Daniel J Gauthier, Erik Bollt, Aaron Griffith, and Wendson AS Barbosa. Next generation reservoir computing. Nat. Commun., 12(1):5564, 2021.
- [10] Junjie Jiang and Ying-Cheng Lai. Model-free prediction of spatiotemporal dynamical systems with recurrent neural networks: Role of network spectral radius. Phys. Rev. Res., 1(3):033056, 2019.
- [11] Kadierdan Kaheman, Steven L Brunton, and J Nathan Kutz. Automatic differentiation to simultaneously identify nonlinear dynamics and extract noise probability distributions from data. Mach. Learn.: Sci. Technol., 3(1):015031, 2022.
- [12] Junjie Jiang and Ying-Cheng Lai. Model-free prediction of spatiotemporal dynamical systems with recurrent neural networks: Role of network spectral radius. Phys. Rev. Res., 1:033056, Oct 2019.
- [13] Deniz Eroglu, Matteo Tanzi, Sebastian van Strien, and Tiago Pereira. Revealing dynamics, communities, and criticality from data. Phys. Rev. X, 10(2):021047, 2020.

- [14] Kevin Course and Prasanth B Nair. State estimation of a physical system with unknown governing equations. Nature, 622(7982):261–267, 2023.
- [15] Oscar Higgott, Thomas C. Bohdanowicz, Aleksander Kubica, Steven T. Flammia, and Earl T. Campbell. Improved decoding of circuit noise and fragile boundaries of tailored surface codes. Phys. Rev. X, 13:031007, Jul 2023.
- [16] Ling-Wei Kong, Hua-Wei Fan, Celso Grebogi, and Ying-Cheng Lai. Machine learning prediction of critical transition and system collapse. Phys. Rev. Res., 3(1):013090, 2021.
- [17] Ling-Wei Kong, Yang Weng, Bryan Glaz, Mulugeta Haile, and Ying-Cheng Lai. Reservoir computing as digital twins for nonlinear dynamical systems. Chaos, 33(3), 2023.
- [18] Zheng-Meng Zhai, Ling-Wei Kong, and Ying-Cheng Lai. Emergence of a resonance in machine learning. Phys. Rev. Res., 5(3):033127, 2023.

REVIEWER COMMENTS

Reviewer #1 (Remarks to the Author):

I thank the authors for making substantial efforts to address my concerns. Overall I am happy and satisfied with the response.

Reviewer #2 (Remarks to the Author):

I commend the authors on an extensive revision with benchmarks that significantly increase the impact of their manuscript with benchmarking.

My only remaining concern is the manner in which α is chosen, which appears to be by ensuring that some random initial conditions call into the known basins of attraction (Fig. 2c, Fig. 4c, Fig. 5b). Is there another way to choose α that does not require certainty about the number or location of the basins? In real-world scenarios, it may not be so obvious from data precisely where the basins may be. Perhaps another measure such as the frequency distribution of the predicted RC outputs? For example, if the data FFT make a clear separation in the low-frequency dynamics and high-frequency noise, one measure of evaluating α would be to see how much of the low-frequency dynamics is captured by the RC. It seems somewhat contrived to select α that guarantees the known number of basins (even if we are ignorant of the true dynamical equations), and then show that the RC does in fact transition between the basins we guaranteed. Hence, a more unbiased measure of learning α that is not directly tied to the explicit convergence towards known basins would be more general.

Another thought that these revisions bring to mind is a 2009 paper "Generating Coherent Patterns of Activity from Chaotic Neural Networks" by Sussillo and Abbott, otherwise known as FORCE learning. FORCE has been shown to be able to better generate time-series data that are not generated from explicit low-dimensional dynamical systems, especially on noisy data. Of course, I am not predicating acceptance on the authors benchmarking against FORCE given their already very thorough revision, but I do invite the authors to at least try out FORCE (and share their results with me if they do!) for science. In my community, FORCE is known to outperform RCs on tasks involving noise and stochasticity where explicit deterministic dynamics are difficult to estimate or do not exist, so it may be a more stringent benchmark.

My formal recommendation is accept with minor revisions (where I would like to see at least some discussion on other ways of choosing α). However, I do believe that the paper would gain significantly more traction with a less contrived way of choosing α , and the results would be much strengthened if the authors found a way to do so.

Reviewer #3 (Remarks to the Author):

I appreciate the authors' effort to revise the paper and to rebut my criticisms. I wish I could be more positive, but the work is not original, the approach undertaken remains questionable, and the main result is problematic.

1. Previous works demonstrated that critical transitions can be anticipated using reservoir computing for nonstationary dynamical systems with time-varying parameters and noise. The types of transitions that have been demonstrated to be predictable by reservoir computing include crisis, amplitude death, explosive synchronization, intermittency, etc. The setting of the present work under review is a bistable system under noise, which is a different case but not more sophisticated than those studied previously. The results reported in the present work are not sufficiently original and significant for a high-impact journal.

2. The whole work was carried out under the presumption that the underlying dynamical system is stationary: there are neither time-varying parameters nor changes in the noise characteristics. The practice of separating noise from the slow dynamics during training and then adding it back during the testing makes no physical sense if the underlying system is nonstationary, where the dynamical behaviors during the training and testing phases can be characteristically different.

3. For the protein folding data, the prediction results are actually not impressive: the histogram of the predicted transition times differs dramatically from the ground truth. The authors claimed reasonable prediction based on one quantity: the average transition time. But even for a long training time (25,000 time steps), there is a large difference between the predicted and actual mean transition time, e.g., Fig. 5e. Figure 5, perhaps the most important figure of the paper, in fact demonstrates that the proposed method does not work, in contrast to authors' claim.

The work is not suitable for Nature Communications.

Responses to Reviewers - NCOMMS-23-43211B

We appreciate all reviewers for their careful reading and constructive comments again, which help improve our manuscript. We are pleased that Reviewer 1 is overall satisfied with the revision. Reviewer 2 also commends our extensive revisions and considers that the impact of the manuscript has significantly increased. Reviewer 3 questions the performance of our method in one example. Accordingly, we have revised the manuscript to eliminate possible misunderstandings. It now becomes more evident that the present method outperforms the previous in accurately predicting noise-induced transitions, making a concrete progress for this wide class of problems.

An overview of the changes is given below, followed by point-to-point responses to the reviewers' comments. The original reports are typed in orange, our responses in black. Please note that the numbers of equations and figures refer to the revised manuscript. The quoted references are listed at the end of the response.

List of main modifications in the revised manuscript

1. Abstract:
 - Revised the presentation to clarify the progress of the present method compared with First-Order, Reduced, and Controlled Error (FORCE) learning.
2. Introduction:
 - Included attempts of applying FORCE learning to noise-induced transitions.
3. Results:
 - Added more details on benchmarking with FORCE learning.
4. Discussion:
 - Discussed that the power spectral density (PSD) of training data can guide the selection of the hyperparameters.
5. Added reference for FORCE learning method [1]. We also cited FORCE learning code library [2].
6. Supplementary Information:
 - Provided more details for the comparison with the FORCE learning including the full-FORCE model and the FORCE-trained spiking neuron model (Supplementary FIGs. 5, 6, 7).
 - Added Supplementary FIG. 8 to demonstrate performance of the present method on the protein folding data.
 - Showed that the PSD can be used to search the optimal α (Supplementary FIG. 9).

Response to Reviewer 1

I thank the authors for making substantial efforts to address my concerns. Overall I am happy and satisfied with the response.

Response:

We thank you for carefully reading our manuscript again. We are glad that you acknowledge our efforts on the revision and are satisfied with the revision.

Response to Reviewer 2

I commend the authors on an extensive revision with benchmarks that significantly increase the impact of their manuscript with benchmarking.

Response:

We thank you for carefully reading our manuscript and for the constructive suggestions to improve the manuscript with more benchmarking. We have followed your suggestions and revised the manuscript accordingly.

My only remaining concern is the manner in which α is chosen, which appears to be by ensuring that some random initial conditions call into the known basins of attraction (Fig. 2c, Fig. 4c, Fig. 5b). Is there another way to choose α that does not require certainty about the number or location of the basins? In real-world scenarios, it may not be so obvious from data precisely where the basins may be. Perhaps another measure such as the frequency distribution of the predicted RC outputs? For example, if the data FFT make a clear separation in the low-frequency dynamics and high-frequency noise, one measure of evaluating α would be to see how much of the low-frequency dynamics is captured by the RC. It seems somewhat contrived to select α that guarantees the known number of basins (even if we are ignorant of the true dynamical equations), and then show that the RC does in fact transition between the basins we guaranteed. Hence, a more unbiased measure of learning α that is not directly tied to the explicit convergence towards known basins would be more general.

Response:

Thank you for raising a crucial point. First, as you noticed, it is helpful to have prior information on the number and locations of the basins, which allows us to evaluate the performance of hyperparameters through the convergence to the correct basins. Indeed, for many datasets, such information can be roughly estimated directly from the time series, such as by segmenting the time series between large jumps and calculating the mean value of each segment. This procedure was used in analyzing the experimental data (Figure 4 of [3]).

We have now developed an alternative protocol by analyzing the power spectral density (PSD) [4], as you suggested, when it is less straightforward to identify the locations of basins directly from data. We find that the PSD of the predicted stochastic time series better matches that of the training data when deterministic dynamics are accurately learned by RC. We observe this when adjusting the time scale α for Example 1 of the main text. Starting with $\alpha = 0.5$ leads to inaccurate learning with mismatched PSDs and transition times (Response FIG. 1(a)). Decreasing α to 0.25 results in better predictions and similar PSDs (Response FIG. 1(b)). Further decreasing α to 0.01 deteriorates model performance and PSD matching (Response FIG. 1(c)). Thus, a better match of PSDs indicates better basin learning, which allows the search of the hyperparameter α without relying on prior information of basins.

Response FIG. 1. **Power spectral density (PSD) can be an indicator for training performance of reservoir computing.** The system is Example 1 of the main text. Discrepancies in PSD are quantified using the root mean square error (RMSE). Green checkmark: good performance. Red cross: poor performance. (a) The trained slow-scale model, when $\alpha = 0.5$. The PSDs of the training (coral line) and the predicted data (green line) exhibit a poor match, with RMSE 0.23. The transition time has larger discrepancies. (b) The case with $\alpha = 0.25$. The PSD of the training data matches that of the prediction, with RMSE 0.18, corresponding to more accurate prediction in transition time. (c) The case with $\alpha = 0.01$. Neither the transition time nor the PSD of the predicted data align with those of the training data.

Another thought that these revisions bring to mind is a 2009 paper "Generating Coherent Patterns of Activity from Chaotic Neural Networks" by Sussillo and Abbott, otherwise known as FORCE learning. FORCE has been shown to be able to better generate time-series data that is not generated from explicit low-dimensional dynamical systems, especially on noisy data. Of course, I am not predicating acceptance on the authors benchmarking against FORCE given their already very thorough revision, but I do invite the authors to at least try out FORCE (and share their results with me if they do!) for science. In my community,

FORCE is known to outperform RCs on tasks involving noise and stochasticity where explicit deterministic dynamics are difficult to estimate or do not exist, so it may be a more stringent benchmark.

Response:

We appreciate the suggestion to explore the FORCE learning [1] for the present examples. We utilize the tension package [2] of FORCE learning and test its different versions, including the full-FORCE model and spiking neuron model. In general, we find that this method is either less accurate in capturing noise-induced transitions, or has a few times longer computational time than our method. The detailed comparisons are below.

Response FIG. 2. **FORCE learning (full-FORCE model) and present method for a one-dimensional bistable system.** The system is Example 1 of the main text. The training data (before the dark dashed line) is $t \in [0, 100]$, with data points plotted every 10 time steps. (a) Prediction in $t \in [100, 200]$ of the full-FORCE model [2]. The predicted data show larger fluctuations compared to the true data. (b) Our prediction has similar fluctuations to the true data, requiring less time than the full-FORCE model.

We first test the full-FORCE model, a basic FORCE learning architecture [1], for the one-dimensional bistable system, Example 1 of the main text. We use the same hyperparameters as that for the task of predicting time series in Figure 4 of [2]. As in Response FIG. 2(a), the prediction after 10 epochs of full-FORCE training exhibits larger fluctuations than the training data, especially at longer time. More importantly, it takes around 10-fold more computational time than the present method (Response FIG. 2(b)).

Response FIG. 3. **FORCE learning (spiking neuron model) for a one-dimensional bistable system with various update intervals.** Update interval: the time steps after which every FORCE update is applied. The system is Example 1 of the main text. Training set: $t \in [0, 100]$, predicting set: $t \in [100, 2000]$. Histograms of transition time for test and predicted data with update intervals of (a) 50, (b) 10, and (c) 1. (d) Prediction in $t \in [100, 200]$ of the spiking neuron model (after the black dashed line), with update interval = 10. Data points are plotted every 10 time steps.

Second, we apply the spiking neuron model, using the same hyperparameter as in Figure 6 of [2]. The key hyperparameter is the update interval, determining how often FORCE updates are applied. We observe that with an interval of 50, as chosen in Figure 6 of [2], the transition time of true and predicted data do not match (Response FIG. 3(a)). We also attempt to search for better values of this hyperparameter. With update interval of 10, the predicted transition time becomes more accurate (Response FIG. 3(b)), with the trajectory shown in Response FIG. 3(d), and is less accurate with a further decrease on update interval to 1 (Response FIG. 3(c)). Despite its match on the transition time for the specific choice of the hyperparameters, the computational time is always a few times more than ours.

Additionally, we apply the spiking neuron model to protein folding data, Example 4 of the main text, with the same training length (T_{train} : 25000, 7500, 6000 time steps). Under the same hyperparameters in Response FIG. 3(b), none of these predictions capture the stochastic transition of this real data (Response FIG. 4). It demonstrates that the spiking neuron model does not give an accurate prediction when using different lengths of training data.

Response FIG. 4. **FORCE learning (spiking neuron model) for the example of protein folding by using various lengths of training data.** The training data is same as the Example 4 of the main text. The update interval = 10, and other hyperparameters are the same as those in Response FIG. 3(b). Black dashed lines represent the boundary between the training and prediction sets. (a) Training length (T_{train}): 25000 time steps; (b) T_{train} : 7500 time steps; (c) T_{train} : 6000 time steps. All the predicted data show no stochastic transitions.

My formal recommendation is accept with minor revisions (where I would like to see at least some discussion on other ways of choosing alpha). However, I do believe that the paper would gain significantly more traction with a less contrived way of choosing alpha, and the results would be much strengthened if the authors found a way to do so.

Response:

We are grateful for your insightful comments regarding the search for the hyperparameter α . Accordingly we have explored using power spectral density to identify a proper value for α . This protocol is particularly useful when stable states are not easily identified from time series data, and strengthens the applicability of the present method to more problems. Thank you again for your constructive feedback and suggestions.

Response to Reviewer 3

I appreciate the authors' effort to revise the paper and to rebut my criticisms. I wish I could be more positive, but the work is not original, the approach undertaken remains questionable, and the main result is problematic.

Response: We thank you for the careful reading again. Based on the responses to each of your comments below, we have now addressed all the remaining concerns. Thus, we believe that the manuscript does report an original approach.

1. Previous works demonstrated that critical transitions can be anticipated using reservoir computing for nonstationary dynamical systems with time-varying parameters and noise. The types of transitions that have been demonstrated to be predictable by reservoir computing include crisis, amplitude death, explosive synchronization, intermittency, etc. The setting of the present work under review is a bistable system under noise, which is a different case but not more sophisticated than those studied previously. The results reported in the present work are not sufficiently original and significant for a high-impact journal.

Response:

We thank you for the comment. Although reservoir computing has been used to study nonstationary dynamical systems with time-varying parameters and noise, another wide class of stochastic phenomena, noise-induced transitions, remains largely unexplored, even for stationary dynamical systems. Unlike the case with parameter drift, these noise-induced transitions represent a crucial area of study that has received considerable attention in the literature [3, 5, 6, 7]. Despite its importance, this phenomenon was not effectively predicted by reservoir computing before, even in the simplest case of a bistable system under noise.

This manuscript fills this gap with a reservoir computing approach, as validated on various examples. We have now also demonstrated that it outperforms the previous state-of-the-art methods, such as SINDy [8] and FORCE learning [1, 2]. Thus, the manuscript makes a significant progress in learning noise-induced transitions by leveraging reservoir computing, and portends the exploration of extending the prevailing machine learning approaches to tackle this wide class of problems.

2. The whole work was carried out under the presumption that the underlying dynamical system is stationary: there are neither time-varying parameters nor changes in the noise characteristics. The practice of separating noise from the slow dynamics during training and then adding it back during the testing makes no physical sense if the underlying system is non-stationary, where the dynamical behaviors during the training and testing phases can be characteristically different.

Response:

Thank you for the point. Although non-stationary systems pose more challenges for the learning task, stationary systems are already difficult to predict and widely encountered. Indeed, as in the pioneering works of reservoir computing [9, 10], the breakthroughs were all

achieved for stationary systems. These important works, as well as many others on reservoir computing [11, 12], have focused on stationary systems. Based on these works, we focus on stationary systems and study noise-induced transitions that were not achieved previously by using reservoir computing alone [13]. Therefore, our study on stationary systems aligns with the scope of existing research and further introduces new methods for a different class of noise-induced phenomena.

3. For the protein folding data, the prediction results are actually not impressive: the histogram of the predicted transition times differs dramatically from the ground truth. The authors claimed reasonable prediction based on one quantity: the average transition time. But even for a long training time (25,000 time steps), there is a large difference between the predicted and actual mean transition time, e.g., Fig. 5e. Figure 5, perhaps the most important figure of the paper, in fact demonstrates that the proposed method does not work, in contrast to authors' claim.

Response:

Response FIG. 5. **Comparison on different methods for predicting transition time for the protein folding data.** Training set lengths: (T_{train}) are (a, c, e) 7500 and (b, d, f) 15000 time steps, and the prediction length is 100000 time steps. Distance: symmetric Kullback-Leibler (KL) divergence between the histograms of transition times. (a, b) Multi-scaling RC: predictions match true data with small distances. (c, d) SINDy-2021 (Threshold = 0.001, Order = 4) [8]: larger distance between the histograms. (e, f) FORCE learning with the same hyperparameters in Response FIG. 4: larger distance.

We respectfully disagree with the reviewer's criticism regarding the accuracy of our predictions for the protein folding data. Our assessment is based on the relative prediction error: for the

upward transition time in FIG. 5(d) of the main text, the relative error is 1.79%, and for the downward transition time in FIG. 5(e) is 16.40%. Indeed, the errors are relatively small when compared with the previous methods, as demonstrated in the next paragraph.

Specifically, for downward transitions (FIG. 5(e) of the main text), we now compare with SINDy-2021 and FORCE learning methods using the same length of training data with 7500 or 15000 time steps. The present method accurately predicts the histogram of transition time with small distances to the histogram from real data (Response FIG. 5(a, b)). Differently, the SINDy-2021 method (Response FIG. 5(c, d)) and the FORCE learning method (Response FIG. 5(e, f)) do not capture the transition time accurately, resulting in more than 5 times larger distance between the histograms. This comparison demonstrates that our method achieves higher accuracy compared to the current state-of-the-art methods.

The work is not suitable for Nature Communications.

Response:

Based on the responses above, we believe that this work can have a wide impact on the field. As another support, the first reviewer is satisfied with the revision, and the second reviewer recommends accepting the manuscript after revisions. Finally, we thank you again for the constructive comments and thoughtful suggestions.

References

- [1] David Sussillo and Larry F Abbott. Generating coherent patterns of activity from chaotic neural networks. *Neuron*, 63(4):544–557, 2009.
- [2] Lu Bin Liu, Attila Losonczy, and Zhenrui Liao. tension: A python package for force learning. *PLOS Comput. Biol.*, 18(12):e1010722, 2022.
- [3] Rafael Tapia-Rojo, Marc Mora, Stephanie Board, Jane Walker, Rajaa Boujemaa-Paterski, Ohad Medalia, and Sergi Garcia-Manyes. Enhanced statistical sampling reveals microscopic complexity in the talin mechanosensor folding energy landscape. *Nat. Phys.*, 19(1):52–60, 2023.
- [4] Pauli Virtanen, Ralf Gommers, Travis E Oliphant, Matt Haberland, Tyler Reddy, David Cournapeau, Evgeni Burovski, Pearu Peterson, Warren Weckesser, Jonathan Bright, et al. Scipy 1.0: fundamental algorithms for scientific computing in python. *Nat. Methods*, 17(3):261–272, 2020.
- [5] Loïc Rondin, Jan Gieseler, Francesco Ricci, Romain Quidant, Christoph Dellago, and Lukas Novotny. Direct measurement of kramers turnover with a levitated nanoparticle. *Nat. Nanotech.*, 12(12):1130–1133, 2017.
- [6] Michael Assaf, Elijah Roberts, and Zaida Luthey-Schulten. Determining the stability of genetic switches: Explicitly accounting for mrna noise. *Phys. Rev. Lett.*, 106:248102, Jun 2011.
- [7] Farshid Jafarpour, Tommaso Biancalani, and Nigel Goldenfeld. Noise-induced mechanism for biological homochirality of early life self-replicators. *Phys. Rev. Lett.*, 115:158101, Oct 2015.
- [8] Alan A Kaptanoglu, Brian M de Silva, Urban Fasel, Kadierdan Kaheman, Andy J Goldschmidt, Jared L Callahan, Charles B Delahunt, Zachary G Nicolaou, Kathleen Champion, Jean-Christophe Loiseau, et al. Pysindy: A comprehensive python package for robust sparse system identification. *arXiv preprint arXiv:2111.08481*, 2021.
- [9] Jaideep Pathak, Brian Hunt, Michelle Girvan, Zhixin Lu, and Edward Ott. Model-free prediction of large spatiotemporally chaotic systems from data: A reservoir computing approach. *Phys. Rev. Lett.*, 120:024102, Jan 2018.
- [10] Daniel J Gauthier, Erik Bollt, Aaron Griffith, and Wendson AS Barbosa. Next generation reservoir computing. *Nat. Commun.*, 12(1):5564, 2021.
- [11] Lufa Shi, Youfang Yan, Hengtong Wang, Shengjun Wang, and Shi-Xian Qu. Predicting chaotic dynamics from incomplete input via reservoir computing with $(D + 1)$ -dimension input and output. *Phys. Rev. E*, 107(5):054209, 2023.
- [12] Huawei Fan, Junjie Jiang, Chun Zhang, Xingang Wang, and Ying-Cheng Lai. Long-term prediction of chaotic systems with machine learning. *Phys. Rev. Res.*, 2(1):012080, 2020.

- [13] Soon Hoe Lim, Ludovico Theo Giorgini, Woosok Moon, and J. S. Wettlaufer. Predicting critical transitions in multiscale dynamical systems using reservoir computing. Chaos, 30(12):123126, 2020.

REVIEWERS' COMMENTS

Reviewer #1 (Remarks to the Author):

I thank the authors for making substantial efforts to address the concerns of the other reviewers and improve the manuscript. Overall I am happy and satisfied with the quality of the manuscript. I am leaning towards a weak accept at this point.

Reviewer #2 (Remarks to the Author):

I commend the authors on a thorough revision including important benchmarking against other methods, and additional detail on the generalizability of the hyperparameter fitting. I recommend acceptance of the manuscript.

Reviewer #3 (Remarks to the Author):

The authors addressed the previous referee comments by presenting new results of comparing the performance of their reservoir-computing scheme with those of two previous methods. I am now convinced that their results on predicting the transition time for protein folding are meaningful and significant. I recommend the paper for Nature Communications.